# Comprehensive analysis of different tumor cell-line produced soluble mediators on the differentiation and functional properties of monocyte-derived dendritic cells

**Sára Burai[1], Ramóna Kovács[1], Tamás Molnár[1], Márta Tóth[1,2,3], Tímea Szendi-Szatmári[4], Viktória Jenei[1], Zsuzsanna Bíró-Debreceni[1], Shlomie Brisco[1], Margit Balázs[5], Attila Bácsi[1,3], Gábor Koncz[1]\*, Anett Mázló[1]\***

1 Department of Immunology, Faculty of Medicine, University of Debrecen, Debrecen, Hungary, 2 Doctoral School of Molecular Cellular and Immune Biology, University of Debrecen, Debrecen, Hungary, 3 ELKH-DE Allergology Research Group, Debrecen, Hungary, 4 Department of Biophysics and Cell Biology, Faculty of Medicine, University of Debrecen, Debrecen, Hungary, 5 Department of Public Health and Epidemiology, Faculty of Medicine, University of Debrecen, Debrecen, Hungary

\* anett.mazlo@gmail.com (AM); konczgb@gmail.com (GK)

**Data Availability Statement:** All relevant data are within the manuscript and its Supporting Information files.

## Abstract

Developing dendritic cells (DCs) from monocytes is a sensitively regulated process. One possible way for cancers to avoid immune recognition and antitumor response is the modulation of DC differentiation. Although several studies are available on the examination of tumor-associated macrophages, a comprehensive analysis focusing on the effects of tumor-formed DCs is not known to date. We provide a comparative analysis of the tumor-edited-monocyte derived DCs differentiated in the presence of adenocarcinomas (MDA, HT29, HeLa)- and primary (WM278, WM983A) or metastatic (WM1617, WM983B) melanomas. The immunomodulatory effect of tumors is mediated at least partly by secreted mediators. We investigated the impact of tumor cell-derived conditioned media on the differentiation of DCs from $CD14^+$ monocytes, sequentially determining the phenotype, cytokine production, phagocytic, and the T cell polarizing capacity of moDCs. We completed our observations by analyzing our data with bioinformatic tools to provide objective correlations between phenotypical and functional properties of different tumor-educated moDCs. The correlation analysis revealed significant differences in the characteristics of adenocarcinomas- or melanomas-edited moDCs. We highlight the functional differences in the properties of moDCs differentiated in the presence of various cancer cell lines. We offer new information and options for the in vitro differentiation protocols of various tumor-conditioned moDCs. Our results confirm that various immunomodulatory properties of different tumor cell lines result in multiple manipulations of DC differentiation.

**Funding:** The work was supported by GINOP-2.3.2-15-2016-00050 project and NTP-NFTÖ-21 and NTP-NFTÖ-22. The work was also supported by National Research, Development and Innovation Office (K125224 and K 142930).

**Competing interests:** The authors have declared that no competing interests exist.

## Introduction

Tumor progression requires active crosstalk between the tumor and its microenvironment. Cancer manipulating immune cells can effectively avoid immune recognition resulting in tumor progression and drug resistance [1]. In the early phase of tumorigenesis, the infiltration of immune cells is associated with the physical destruction of the tumor cells and improved clinical outcome. However, long-lasting immunological pressure results in the development of tumor immune escape mechanisms, including reprogramming of immune cells. This immune-editing can be performed by releasing cytokines, chemokines, exosomes, apoptotic bodies from tumor cells, or other recruited and manipulated cells [2]. Reprogramming innate immunity, controlled/modulated development, and maturation of monocytes and monocyte-derived cells are closely related to successful tumor growth and spread [3, 4].

To reveal the factors and relationship between the highly plastic cancer-associated myeloid cells and malignant tumors seems crucial. Monocytes are highly plastic myeloid-derived cells with the potential to differentiate into macrophages or dendritic cells (DC) [5]. While numerous studies focus on generating tumor-associated macrophage (TAM) subtypes, exploring the differentiation of tumor-related dendritic cell subpopulations is a less prominent area. Although the formation of tumor-promoting DC subsets is also forced during tumorigenesis [2]; tumor-infiltrating DCs have been described in multiple histologies [6].

DCs represent an essential cellular component in communication between the innate and adaptive immune systems. They are highly plastic cells; thus, their differentiation into tolerogenic or immunogenic phenotypes is controlled by the surrounding environment. Their unique function is collecting and processing antigens, transporting them to secondary lymphoid organs, and cross-presenting intracellular peptides to MHCI-restricted cytotoxic T cells, activating naive $CD8^+$ cells. DC-mediated antigen transport and expression in lymph nodes, are also required for helper T cell activation and polarization. In addition, DCs regulate the functional programming of both cytotoxic and helper effector T lymphocytes in peripheral tissues [7]. Although all major subsets of DCs are critical components of anti-tumor immunity, Irf8- and Batf3-dependent cDC1 ($Xcr1^+CD103^+$) lineage is specialized for antigen cross-presentation with MHC class I (MHCI) molecules to activate $CD8^+$ T cells [8]. Through their T-cell regulatory functions, DCs in different localization (tumor-bed-/ tumor-draining lymph node-/ distant metastatic sites) may play a prominent role in tumor regulation and influencing responses to immunotherapy, especially for checkpoint therapies [9]. However, their plasticity makes these cells highly vulnerable to tumor-mediated manipulations. Despite progress in understanding DC subsets, it remains unclear whether various subtypes of tolerogenic DC populations develop in the environment of different tumor types. There are studies characterizing the tumor-conditioned DCs; melanoma cells induce the enhanced expression of CD80, CD86, MHC class I, and MHC class II molecules on immature DC. Expression of E-cadherin and strong upregulation of CD15 also be detected and the absence of CD83 expression.

Additionally, DC mobility and antigen presentation are also impaired by tumor cells [10]. In a non-small cell lung cancer study, the CD11c high tumor-infiltrating dendritic cell (TIDC) subset expressed a low level of co-stimulatory molecules but a high level of inhibitory programmed death-1 ligand (PD-L1). It exhibited a poor response to TLR stimulation [11]. TIDCs tend to trigger the expansion of $FOXP3^+$ regulatory T cells supporting the generation of immune tolerance [12].

Because access to patients' tumor-educated cells is limited, the optimization of in vitro differentiation protocols for the generation of tumor-supporting myeloid cells appears to be a forward-looking opportunity to test new intervention strategies. Additionally, because of the poor predictability of animal models regarding the human immune responses and the 3Rs

guidelines (reduction, refinement, and replacement of experimental animals) of the EU Directive 2010/63/EU, in higher number and more efficient in vitro systems offered to be created [13]. An increasing number of methods have already been published to generate tumor-educated monocyte-derived cells underlying the major effect of tumors on the differentiation of myeloid cells. MDA-MB231, HT29, and HeLa are extensively studied human adenocarcinoma cell lines from breast, colon, and cervical cancer. Conditioned medium collected from all these three cell lines was able to shift macrophage differentiation into TAM direction [14–16]. Primary WM278 and metastatic WM1617 melanoma cell lines originate from the same individual, and primary WM983A is derived from the same patient as the metastatic WM983B cell lines. Moreover, micro-environmental crosstalk directs immunosuppressive M2-like macrophage differentiation in human melanoma models, investigated with WM278, WM983A primary and WM1617, and WM983B metastatic melanoma cell lines [13].

To date, a comprehensive analysis focusing on the effects of the different tumor cell line-derived condition media (TU-CM) on the phenotype and function of the DCs differentiated in the presence of IL-4 and GM-CSF is unknown yet. Our study compares the effect of adenocarcinoma- and melanoma-derived cell lines on *in vitro* differentiation of DCs. Our results may serve as a basis for *in vitro* investigation of different tumor-promoting DCs, which offer an easy-to-use assay system to study the therapeutic manipulation of tumor-induced DC differentiation.

## Material and methods

### Human moDC cultures

Heparinized leukocyte-enriched buffy coats were obtained from healthy blood donors drawn at the Regional Blood Center of the Hungarian National Blood Transfusion Service (Debrecen, Hungary) in accordance with the written approval of the Director of the National Blood Transfusion Service and the Regional and Institutional Research Ethical Committee of the University of Debrecen, Faculty of Medicine (Debrecen, Hungary). Written, informed consent was obtained from the blood donors prior to blood donation; their data were processed and stored according to the directives of the European Union.

Peripheral blood mononuclear cells (PBMCs) were separated from buffy coats by Ficoll-Paque Plus (Amersham Biosciences) gradient centrifugation. According to the manufacturer's protocol, monocytes were purified from PBMCs by positive selection using immunomagnetic anti-CD14-conjugated microbeads (Miltenyi Biotec). After separation on a VarioMACS magnet, 96–99% of the cells were shown to be CD14$^+$ monocytes, as measured by flow cytometry. Isolated monocytes were cultured for four days in 6-well tissue culture plates at a density of 1.5 x $10^6$ cells/ml in either RPMI (Sigma-Aldrich) or TU-CM media, both supplemented with 10% FCS (Gibco), 1% antibiotic/antimycotic solution (Hyclone) in the presence of 100 ng/ml IL-4 (PeproTech EC) and 80 ng/ml GM-CSF (Gentaur Molecular Products).

### Generation of monocyte-derived dexamethasone DCs

Dexamethasone DCs (dexDCs) were generated by culturing isolated monocytes at 1.5x$10^6$ cells/ml concentration in RPMI supplemented by 10% FCS, 1% antibiotic/antimycotic solution in the presence of 100 ng/ml IL-4, 80 ng/ml GM-CSF and 0.25 μM dexamethasone (Sigma-Aldrich).

**Cell lines.** MDA-MB231 (human breast adenocarcinoma), HeLa (human cervical cancer), HT29 (colon cancer), WM278 (primary melanoma), WM1617 (the metastatic pair of WM278), WM983A (primary melanoma), and WM983B (the metastatic pair of WM983A)

cell lines were cultured in RPMI supplemented with 10% FCS and 1% antibiotic/antimycotic solution.

## Generation of tumor cell-line conditioned media

MDA-MB231 (human breast adenocarcinoma), HeLa (human cervical cancer), HT29 (colon cancer), WM278 (primary melanoma), WM1617 (the metastatic pair of WM278), WM983A (primary melanoma), and WM983B (the metastatic pair of WM983A) cell lines were cultured in RPMI supplemented with 10% FCS and 1% antibiotic/antimycotic solution. Supernatants were removed from the tumor cell-line cultures, adherent cultures were washed, and media were changed to fresh ones (RPMI supplemented with 10% FCS and 1% antibiotic/antimycotic solution). The cultures were incubated for 48 hours. TU-CMs were collected and centrifuged for 5 minutes at 3000 rpm at room temperature. The ratio of fresh and TU-CM RPMI was 1: 1 during the monocyte differentiation process.

## Separation of peripheral blood lymphocytes

PBMCs were separated from buffy coats by Ficoll-Paque Plus gradient centrifugation. According to the manufacturer's protocol, monocytes were purified from PBMCs by positive selection using immunomagnetic anti-CD14-conjugated microbeads. At the end of this process, the monocyte-depleted peripheral blood lymphocytes (PBL) were frozen immediately after the separation. The cells were used for the T cell polarization assay as autologous PBL, including helper and cytotoxic T cells. On day 4 of monocyte differentiation, monocyte-derived cells were washed, and they were cocultured with freshly thawed autologous PBL ($1\times10^5$ DC: $1\times10^6$ PBL)

## DC-T cell co-cultures

Autologous peripheral blood lymphocytes (PBLs) were used for the T cell polarization assay. Monocytes were purified from PBMCs, as written previously. Control or TU-CM-moDCs were counted, washed, and co-cultured with autologous PBLs for three (Th1, Tc), five (Th2, Th17), or nine (Treg) days at a ratio of 1:10 (moDCs: T-cells) in RPMI-1640 medium supplemented by 10% FCS, 1% antibiotic/antimycotic solution.

## Flow cytometry

Phenotyping of resting and TU-CM-moDCs was performed by flow cytometry using anti-human CD14-fluorescein isothiocyanate (FITC), CD209/DC-SIGN-phycoerythrin (PE), CD1a-FITC, CD1d-Peridinin-Chlorophyll-Protein (PerCP), CD80-FITC, CD86-PE, PD-L1-PE, CD163 –PE, CD206 –allophycocyanin (APC) (all from BioLegend), CTLA-4-PE (Sony Biotechnology Inc.), HLA-DR-FITC (BD Biosciences). Cell viability was assessed by 7-aminoactinomycin-D (7-AAD; 10 μg/ml; Sigma–Aldrich) staining. Samples were stained for 10 minutes with 7-AAD immediately before flow cytometric analysis. Fluorescence intensities were measured by Novocyte2000R Flow Cytometer (Agilent (Acea) Biosciences Inc., USA), and data were analyzed by the FlowJo v X.0.7 software (Tree Star).

## Measurement of the cytokines by ELISA or flow cytometry

Supernatants of moDCs and TU-CM-moDCs were harvested four days after monocyte separation or 24 hours after the stimulation of moDCs. The concentration of IL-6, IL-10, TNFα cytokines, and chemokine IL-8 was measured and validated using OptEIA kits (BD Biosciences) following the manufacturer's instructions.

To determine which T-lymphocyte populations are responsible for the cytokine production, the T cells were stimulated with 1 μg/ml ionomycin and 20 ng/ml phorbol-myristic acetate (PMA) for 4 hours. The vesicular transport was inhibited by BD GolgiStop™ protein transport inhibitor (BD Biosciences) after the activation. The cells were then labeled with anti-human CD3-FITC, anti-human CD8-PE or CD4-PerCP, and anti-human CD25-PE antibodies (all from BioLegend). The samples were then fixed and permeabilized by BD Cytofix/Cytoperm™ Plus Fixation/Permeabilization Kit (BD Biosciences) and labeled again with anti-human CD8-PE or CD4-PerCP as well as with anti-human IFNγ-APC (BD Biosciences), anti-human IL-4-PE (R&D Systems), anti-human IL-10-Alexa Fluor 488, anti-human IL-17-PE (BioLegend). T cell staining panel is shown in S1 Fig. Fluorescence intensities were measured by Novocyte2000R Flow Cytometer (Agilent (Acea) Biosciences Inc., USA), and data were analyzed by the FlowJo v X.0.7 software (Tree Star).

## The mCherry-expressing *Lactobacillus casei* BL23 strain and its growth conditions

The fluorescent *L. casei* strain was generated by transforming wild-type bacteria with the red fluorescent mCherry protein encoding pTS-mCherry plasmid (pTS-mCherry was provided by Marie-Pierre Chapot-Chartier, INRAE Centre Jouy-en-Josas, Jouy-en-Josas, France). Transformed bacteria were routinely inoculated from frozen glycerol stocks on an MRS agar plate containing 5 μg/ml erythromycin for clonal selection. The plate was cultured for 48 h at 37˚C to obtain single colonies, which were then added to MRS broth supplemented with 5 μg/ml erythromycin and incubated overnight at 37˚C. The optical density of 1 at 600 nm corresponds to $2.5 \times 10^8$ bacterial cells/ml.

## Phagocytosis assay

Monocytes were cultured for four days in 6-well tissue culture plates at a density of 1.5 x 106 cells/ml in conditioned TU-CM media or RPMI-1640 medium. For the phagocytosis assay, DCs were co-cultured with mCherry-expressing *L. casei* bacteria at a ratio of 1:4 for 4 hours. Fluorescence intensities were measured by Novocyte 3000RYB Flow Cytometer (Agilent (Acea) Biosciences Inc., USA), and data were analyzed by the FlowJo v X.0.7 software (Tree Star).

## Confocal microscopy

To examine the localization of captured bacteria after the phagocytoses, the moDCs were subjected to confocal microscopic analysis. Zeiss LSM 880 confocal laser scanning microscope (Carl Zeiss, Oberkochen, Germany) equipped with 40x water immersion objective (NA 1.2) was used to image the samples. The following lasers were used as the excitation source: 405 nm diode laser for DAPI; 488 nm line of an Argon ion laser for FITC; 543 nm He-Ne laser for mCherry. Fluorescence emissions of DAPI and FITC were detected in the wavelength range of 410–485 and 490–610 nm, respectively, while the detection of mCherry was performed with a 575–695 nm band-pass filter. Z-stack images were collected at 1 mm intervals from the bottom to the top of the cells. The pictures were analyzed by the ImageJ 1.51 software (National Institute of Health, USA).

## Bioinformatical analyses

R (version 4.1.3) [17] and RStudio (version 1.4.1717) were used for bioinformatical analyses. The dataset contained the mean of marker expression, cytokine secretion, T-cell polarizing,

and phagocytosis data. Correlation matrices were performed by *R base* package (v 4.1.3) with *cor* function for adenocarcinoma and melanoma cell lines-conditioned moDCs separately. Correlograms from correlation matrices were plotted by *corrplot* (v 0.92) package [18] with *corrplot* function using the following parameters: type = upper, order = original. P-value matrix and confidence intervals matrix were added using the *cor.mtest* function with confidence level 0.95 of the *corrplot* package. Significance levels were defined as *P < 0.05; **P < 0.01; ***P < 0.001. Mixed correlograms were performed by *corrplot.mixed* function of the *corrplot* package with the parameters lower = number, upper = circle, order = original. Heatmap was generated from the raw data by *the pheatmap* function of the *pheatmap* (v 1.0.12) library [19] using the scale = column parameter. Principal Component Analysis (PCA) was made by the *PCA* function of the *FactoMiner* package (v 2.4) [20] using the default values of scale.unit, ncp and graph parameters from raw data with or without controls (RPMI and dexDC) separately. Results of the PCA were visualized by *factoextra* package (v 1.0.7) [21] with *fviz_pca_ind* function with default parameters.

## Statistical analysis

One-way ANOVA followed by Tukey's post hoc test was used for comparisons for more than two groups. The results were expressed as mean +SD. Analyses were performed using Excel (Microsoft Corporation) and GraphPad Prism Version 6.0 (GraphPad Software Inc.) software. Differences were considered to be statistically significant at P < 0.05. Significance was indicated as *P < 0.05; **P < 0.01; ***P < 0.005; ****P < 0.0001.

## Results

### TU-CMs modify the expression of dendritic cell differentiation markers

To get insight into how the presence of TU-CMs affects the phenotypic characteristics of moDC differentiated *in vitro*, we monitored the expression of cell surface molecules by flow cytometry. Characteristic differentiation markers of moDCs, including CD14 [22], DC-SIGN [23], and group 1 CD1 family members [22], were compared on moDCs conditioned by different tumor cell lines (adenocarcinoma cell lines; MDA, HT29, HeLa, primer melanoma cell lines; WM278, WM983A, and metastatic melanoma cell lines; WM1617, WM983B). DCs treated with dexamethasone were used as controls for tolerogenic DCs, according to previously published protocols [24–27]. None of these treatments modified DC viability (S2 Fig), and the conditioning of monocytes with TU-CMs during the differentiation process did not affect the autofluorescence of the cells (S3 Fig). HT29-, WM278-CM-moDCs and according to the literature [28], DexDCs showed significantly higher CD14 expression than the control moDCs (Fig 1A). The DC-SIGN (CD209) molecule's cell surface expression was significantly down-regulated on WM983A and WM983B-exposed cells compared to the RPMI-moDCs. The reciprocal changes in the expression of CD14 and DC-SIGN are associated with the loss of epigenetic activation at the CD14 locus, but the acquisition of the same at the DC-SIGN locus [29]. Because of the strict co-regulation of these markers, we show the quadrant statistic of the expression of CD14 and DC-SIGN (Fig 1A). As the dexamethasone-treated cells, every tumor cell line-derived-CM induced the generation of CD14 and DC-SIGN double-positive moDCs. However, significant differences were observed when the effects of different TU-CMs were compared on the appearance of these two markers. HT29, WM278, and WM983B-conditioned moDCs showed a higher frequency of CD14 and CD209 co-expression. Interestingly, significantly more double-positive DCs developed after treatment with metastatic melanoma WM983B-CM than after differentiation in the presence of primary WM983A melanoma-CM. Double negative cells were detected in moDC cultures, where the monocytes differentiated in

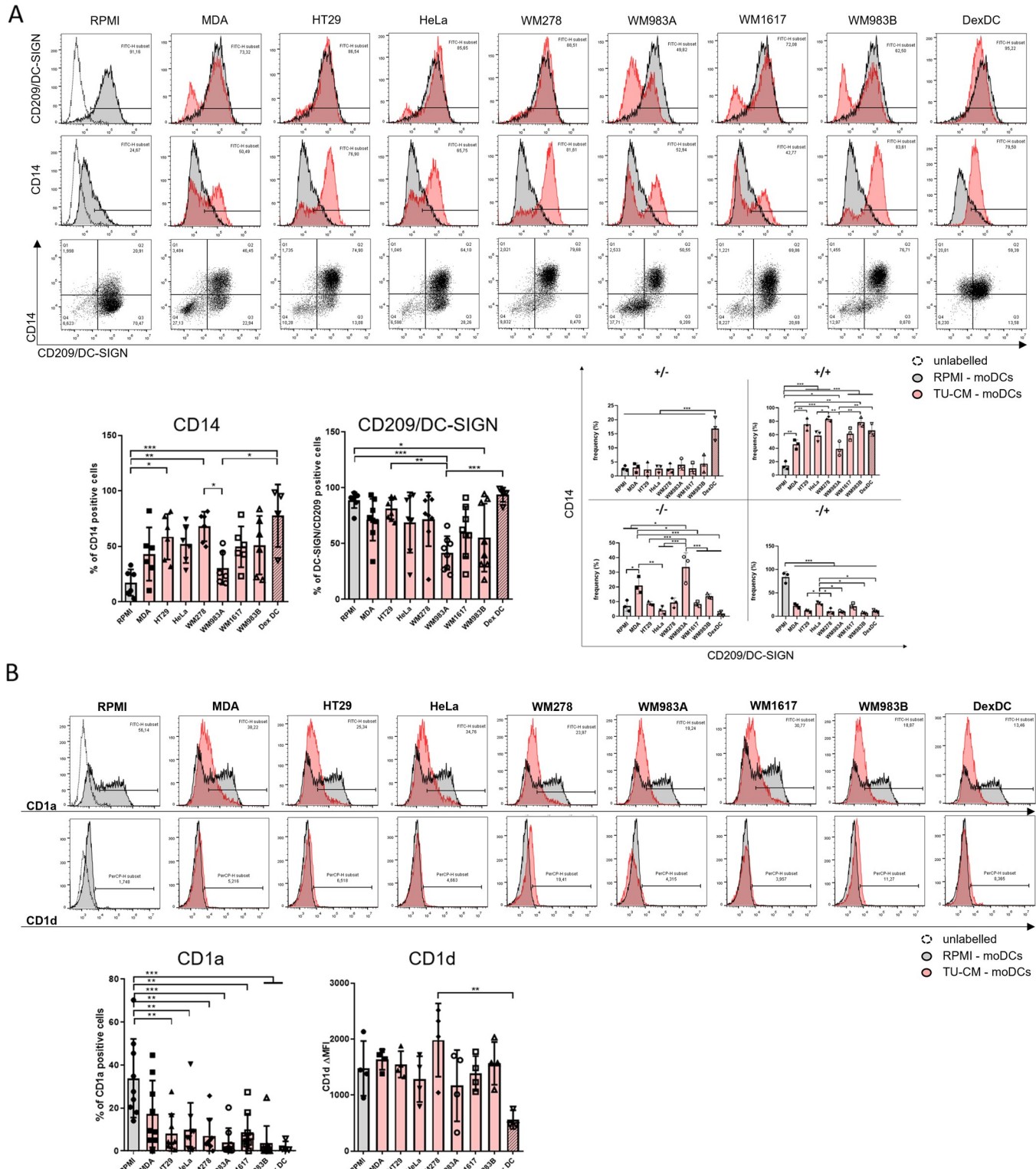

**Fig 1. TU-CMs modify the phenotype of dendritic cells.** CD14[+] monocytes were cultured with 100 ng/ml IL-4, 80 ng/ml GM-CSF ± MDA-MB231-, HeLa-, HT29-, WM278-, WM1617-, WM983A- or WM983B-CM for four days. On day 4, the cell surface expression of (A) CD14 and DC-SIGN or (B) CD1a and CD1d were analyzed by flow cytometry. Figure A and B show the MFI (median fluorescence intensity) plus SD or the mean plus SD values of the populations positive for the measured surface molecules calculated from four independent experiments. Histograms or dot plots show one of the four independent experiments. Significance is indicated by $^*P < 0.05$, $^{**}P < 0.01$, $^{***}P < 0.001$.

the supernatant of MDA or WM983A cell lines. In the case of DCs differentiated in HeLa-CM, significantly more CD14⁻CD209⁺ cells were identified than in the cultures generated in HT29, WM278, WM983A, WM983B, or dexamethasone (Fig 1A). Our results suggest that tumor cell supernatants and dexamethasone alter the epigenetic regulation of moDCs' CD14 and DC-SIGN expression differently.

Exposure to any TU-CM significantly reduced the cell surface expression of CD1a glyco-lipid receptor on monocyte-derived cells (Fig 1B). In general, the expression of CD1d on moDCs was not significantly modulated by tumor cells, while a significant difference was observed between the WM278 and DexDC groups. However, both CD1a and CD1d molecules involved in NKT cell activation, the expression of CD1a is strongly regulated during the differentiation process of moDCs, determining their inflammatory or anti-inflammatory functional features [30]. The inhibited appearance of CD1a by tumor cells or dexamethasone promotes the generation of anti-inflammatory moDCs. Considering the alternative hypothesis, the phenotype and properties of the monocyte-derived cells differentiated in the presence of conditioned media are consistent with the monocyte-derived macrophages. We performed a further phenotypic analysis of monocyte-derived cells differentiated in the presence of TU-CMs (S4 Fig). We found that WM983B significantly up-regulated the expression of CD163, in which condition the expression of DC-SIGN and CD1a was significantly silenced. However, only a small percentage of the cells expressed the CD163 molecule. In conclusion, WM983B-produced-CM promotes the differentiation of CD1a-DC-SIGNmed monocyte-derived cells with a small population of CD163⁺ macrophage-like cells. However, the expression of CD206 was significantly diminished on the surface of WM983B-CM-conditioned monocyte-derived cells compared to the control moDCs (RPMI). Additionally, in line with the literature, differentiation of monocytes in the presence of IL-4, GM-CSF, and dexamethasone induces the increased expression of CD163 [31]. In conclusion, WM983B-produced-CM promotes the differentiation of CD1a⁻DC-SIGN^med monocyte-derived cells with a small population of CD163⁺ macrophage-like cells.

## TU-CMs alter the cell surface expression of molecules involved in T cell activation on DCs

We examined the effect of TU-CMs on the surface expression of MHC class I (HLA-ABC), MHC class II (HLA-DR), CD86 (B7-2), and programmed death-ligand 1 (PD-L1 / B7-H1), which are critical components in the regulation of T cell activation. While dexamethasone treatment increased MHCI and MHCII expression on DCs, none of the DCs treated by supernatant of tumor cell lines modified MHCI or MHCII expression significantly. However, WM983A and WM983B cells tended to express lower amounts of MHC (Fig 2 and S5 Fig). According to published data, DexDCs expressed only minimal amounts of co-stimulatory CD86 [32]. Still, DCs in the presence of TU-CMs did not modify or only slightly increase CD86 expression compared to control DCs. The frequency of positive/highly positive cells are presented in Fig 2 and MFI values in S5 Fig. We also examined the level of the surface co-inhibitory molecule PD-L1. Compared to control DCs, none of the tumor supernatant-treated DCs showed a significant change in PD-L1 expression, but there were large differences in TU-CM-treated cells compared to each other. We found that compared to other TU-CM-treated DCs, PD-L1 expression in WM983A and partly in WM983B cells was downregulated, similar to that observed in DexDC (Fig 2 and S5 Fig). These results indicate that tumor supernatants do not intensively modify the expression of T cell regulatory molecules on DCs. In this respect, the effect of TU-CM does not mimic the effect of tolerogenic dexamethasone treatment.

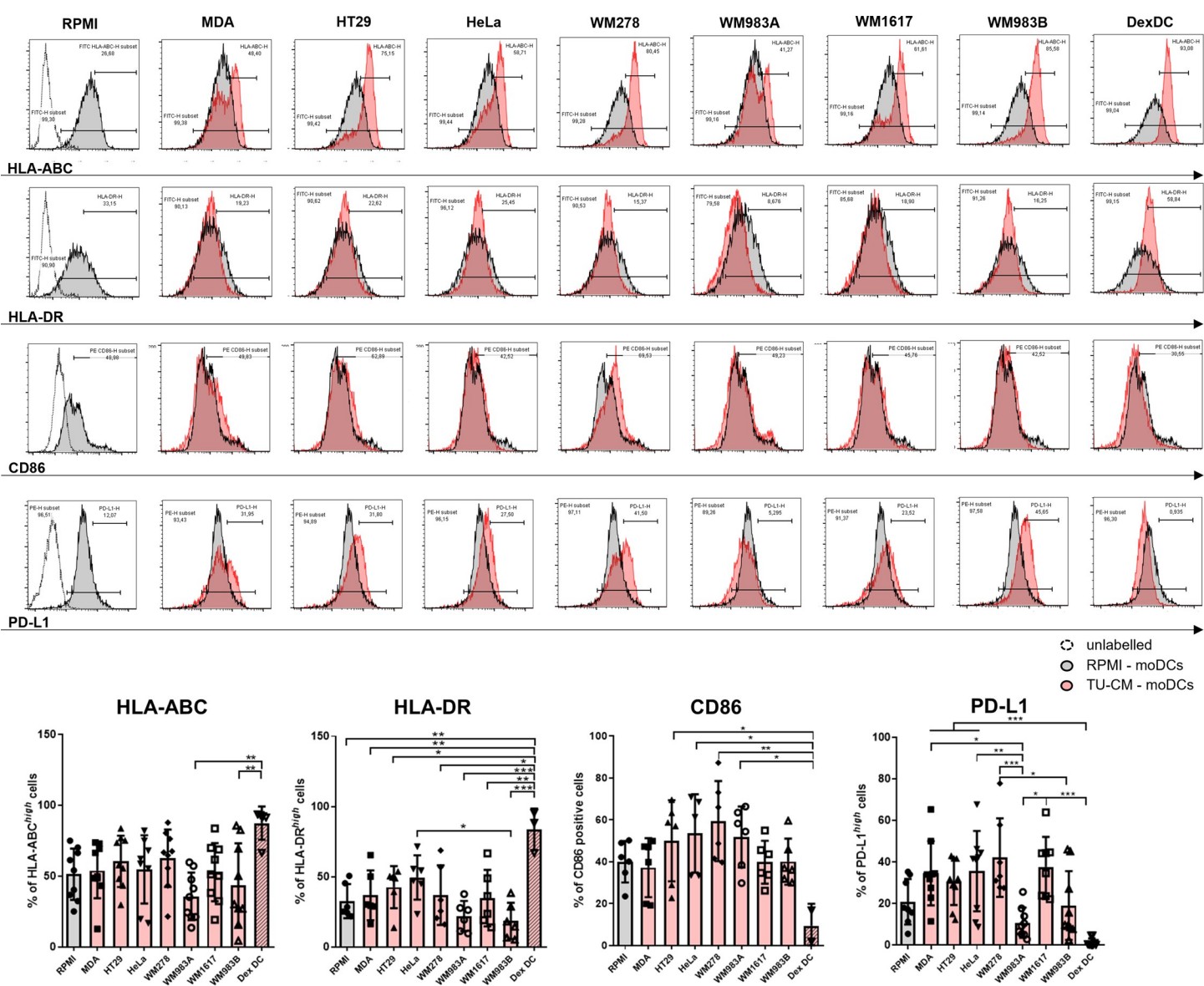

**Fig 2. TU-CMS alters the cell surface expression of molecules involved in conventional T cell activation.** CD14[+] monocytes were cultured with 100 ng/ml IL-4, 80 ng/ml GM-CSF ± MDA-MB231-, HeLa-, HT29-, WM278-, WM1617-, WM983A- or WM983B-CM for four days. On day 4, the cell surface expression of HLA-ABC, HLA-DR, CD86, and PD-L1 was analyzed by flow cytometryThe figure shows the mean plus SD values of the populations positive for the measured surface molecules calculated from five independent experiments. Histograms or dot plots show one of the five independent experiments. Significance is indicated by *P < 0.05, **P < 0.01, ***P < 0.001.

### TU-CMs modulate the cytokine and chemokine production of moDCs

Cytokine production, phagocytic activity, and T cell polarization ability of TU-CM-differentiated DCs were determined for functional characterization. Inflammatory (TNF-α, IL-6), immunosuppressive (IL-10) cytokine production, and chemokine (IL-8) secretion were detected in the supernatant of differentiated moDCs. MDA cells and melanoma cell lines, except WM1617, produced considerable amounts of IL-8. IL-6 production by MDA cells was also significant (Fig 3B). In addition to their direct cytokine secretion, tumor cells also affected DC-mediated cytokine production. In line with published data, dexamethasone treatment

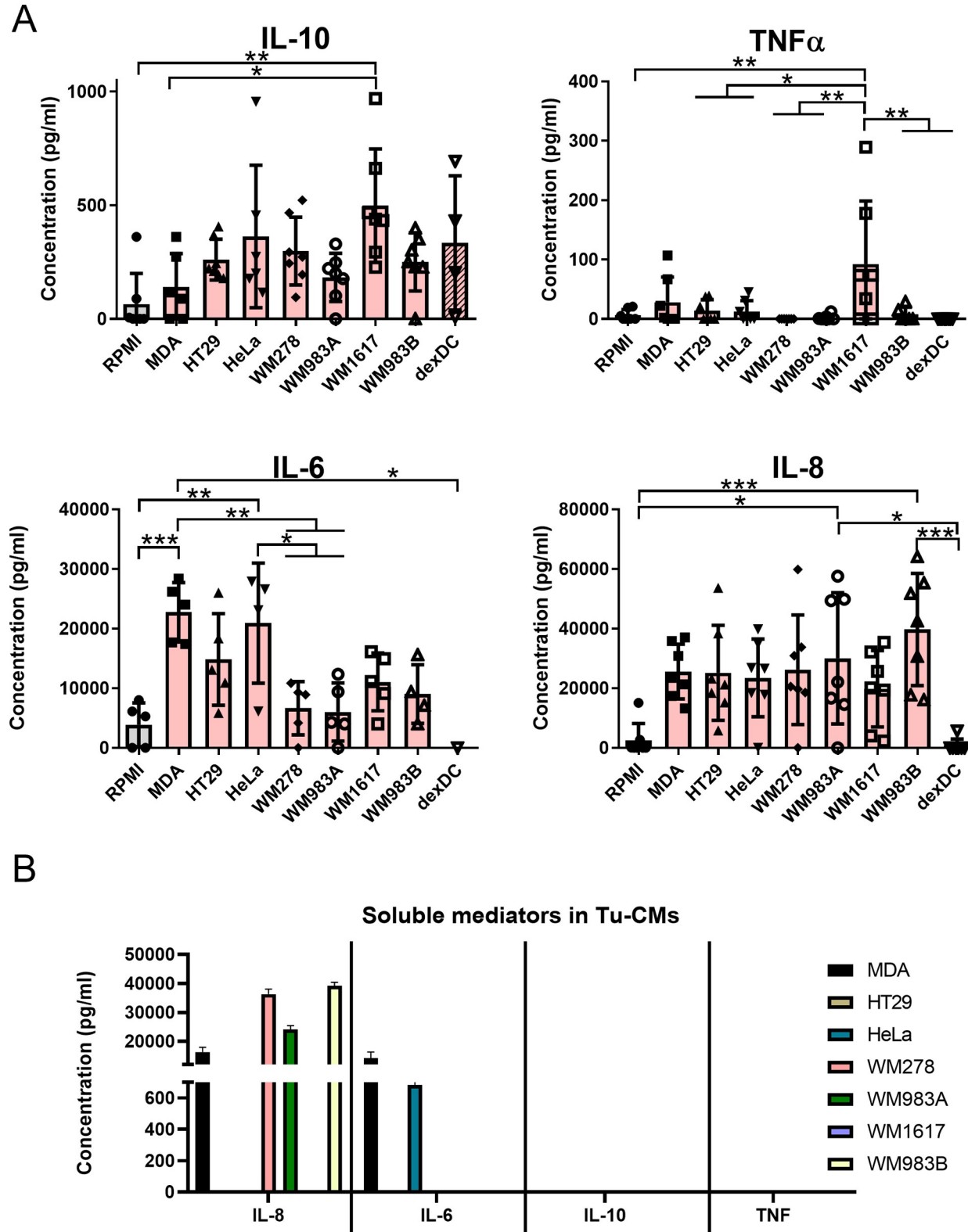

**Fig 3. TU-CMs modulate the cytokine and chemokine production of moDCs.** (A) CD14[+] monocytes were cultured with 100 ng/ml IL-4, 80 ng/ml GM-CSF ± MDA-MB231-, HeLa-, HT29-, WM278-, WM1617-, WM983A- or WM983B-CM for four days. The concentration of cytokines secreted by the moDCs or (B) the cytokine/chemokine content and concentration of TU-CM conditioned media were detected by ELISA. Mean

plus SD of relative cytokine levels and concentrations were calculated from at least four independent experiments. Significance is indicated by
*P < 0.05, **P < 0.01, ***P < 0.001.

decreased inflammatory cytokine production but induced high levels of IL-10 secretion [25]. Typically, the IL-6 production of adenocarcinoma cell lines (especially MDA and HeLa) was significantly higher than IL-6 secretion of the non-treated or melanoma supernatant-treated DCs.

In contrast to IL-6 production, treatment with tumor cell-derived supernatants generally did not modify the TNFα secretion by moDCs, but the supernatant of WM1617 metastatic melanoma cell line induced significantly increased TNFα production (Fig 3A and S6B Fig). An overall increase in IL-8 cytokine production was observed in all TU-CM-treated DCs compared to the control or dexamethasone-treated groups. The production of the immunosuppressive cytokine IL-10 was significantly upregulated in the presence of WM1617 metastatic melanoma cell line-CM and slightly increased after all tumor supernatants and dexamethasone treatment (Fig 3A and S6A Fig). In general, tumor cells significantly modulated the cytokine production of DCs. IL-6 production by moDCs also differed in the case of adenocarcinoma- and melanoma-induced regulation.

## TU-CMs alter the phagocytic capacity of monocyte-derived cells

The phagocyte function of moDCs was examined by the uptake of mCherry-expressing *Lactobacillus casei* BL23. Confocal microscopic images show that mCherry-positive bacteria were taken up by moDCs with high efficiency (Fig 4A). Consistent with the literature, dexamethasone treatment increased phagocyte activity [25–30, 32, 33] (Fig 4B). A significant difference can be detected in the phagocytic activity of the control moDCs and WM278, WM1617, or DexDCs, as well as between the HT29- and WM278-CM-conditioned monocyte-derived cells. These results may indicate that certain tumors can alter the phagocytic capacity of the DCs.

## Tumor cell line-educated moDCs polarize T cells in different ways

Based on our results, TU-CMs did not strongly modify the co-stimulatory potential of moDCs (Fig 2). To explore the T cell polarizing capacity of moDCs, we examined the presence of the intracellular IFNγ (Th1 or cytotoxic T cells—Tc), IL-4 (Th2), IL-17 (Th17), or IL-10 cytokine levels along with CD25 (regulatory T cells—Treg) expression. Since we found no differences in the Th1, Th2, and Th17 activatory potential between differently generated tumor-educated moDCs (Fig 5), we focused on the two most critical T-cell types in the tumor microenvironment, CD8[+] cytotoxic and CD4[+] regulatory T-cells (Fig 5). Consistent with published data [34], DexDCs inhibited the IFNγ production of cytotoxic T lymphocytes (Fig 5). Similarly, we have detected inhibited cytotoxic T cell activation in the presence of WM983B metastatic melanoma cell line-educated moDCs; in contrast, its primary counterpart induced the activation of IFNγ-producing CD8[+] T cells (Fig 5). Additionally, we measured the intracellular level of IL-10 in the CD4[+]CD25[+] subpopulation (Fig 5). Interestingly, in opposed to the WM1617 metastatic melanoma cell line, WM983B-educated moDCs were prone to enhance the IL-10 production of CD4[+]CD25[+] lymphocytes. We detected significant differences in the IL-10-producing lymphocyte activation capacity between metastatic melanoma cell lines and MDA/ HT29 adenocarcinoma cell lines. Additionally, HT29 had a significantly lower capacity to induce IL-10 production by CD4[+]CD25[+] lymphocytes (Fig 5) than the HeLa adenocarcinoma cell line. According to our data, the T-cell polarization ability of DCs altered by different

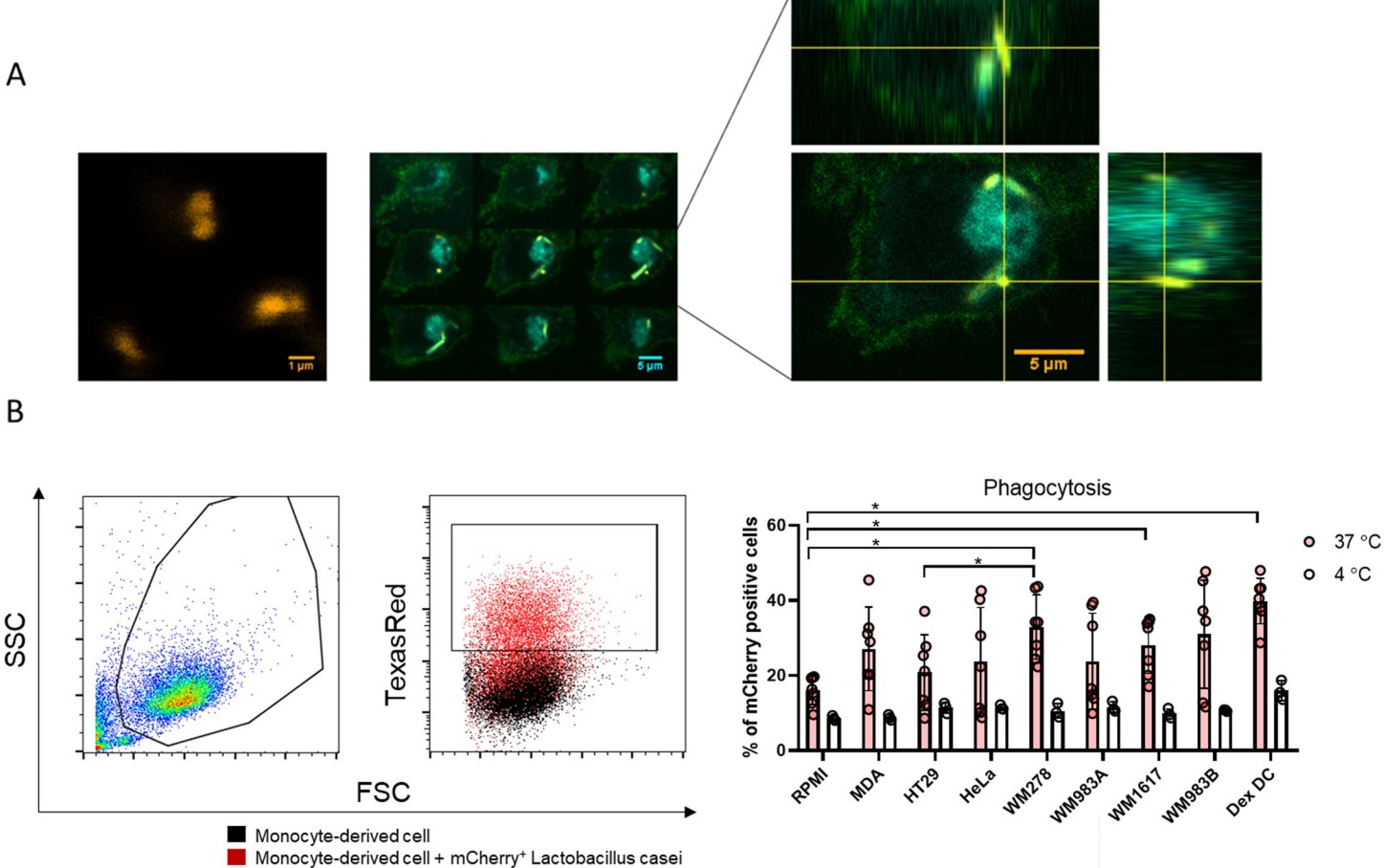

**Fig 4. TU-CMs alter the phagocytic capacity of monocyte-derived cells.** (A) CD14+ monocytes were cultured with 100 ng/ml IL-4, 80 ng/ml GM-CSF ± MDA-MB231-, HeLa-, HT29-, WM278-, WM1617-, WM983A- or WM983B-CM for four days. The phagocytic capacity of tumor cell line-educated moDCs was measured by the uptake of mCherry-expressing *Lactobacillus casei*. Microscopic images show DCs labeled by FITC conjugated HLA antigens (green), nuclei were visualized with DAPI (cyan) and mCherry-expressing *Lactobacillus casei* (yellow). The scale bar is 1 μm (left picture) or 5 μm (middle and right picture). In the middle Z-stack images of the cells, Z-stack images were collected in 1 μm sections from the bottom to the top of the cell, scale bar is 5 μm. In the figure on the right is the orthogonal view of Z-stack images. On the top is Z-projection in the X-Z direction, on the right is the Z-projection in the Y-Z direction. The yellow lines indicate the Z-depth of the optical slice. The yellow lines show the orthogonal planes of the X–Z and Y–Z projection, respectively. (B) *In vitro* differentiated four-day monocyte-derived cells were coincubated with mCherry-labeled *L. casei* for 4 h at a ratio of 1:10 at 37˚C and 4˚C as a control. Phagocytic activity of moDCs was measured by detecting mCherry fluorescent protein-positive moDCs with flow cytometry after 4h. The diagram shows the mean plus SD of seven (37˚C) or three (4˚C) independent experiments. Boxplot shows one representative sample from the seven independent experiments. Significance is indicated by *P < 0.05.

tumor cell lines promotes the development of IFNγ or IL-10-producing T cells, but different tumors induce diverse responses.

## Principal component analysis of the phenotypical and functional properties of moDCs differentiated in tumor cell lines-derived conditioned media

To examine the overall effect of dexamethasone or the cancer cell lines-derived supernatants on the moDCs, principal component (PCA) and correlation analysis were performed by using the data from flow cytometry and ELISA (Fig 6, S6 and S7 Figs). The analysis revealed that tumor cells-educated moDCs were well distinguished from dexamethasone-treated or control (RPMI) DCs (S6A and S6B Fig). Bioinformatic analysis was performed to reveal differences in the strategy of melanomas and adenocarcinoma cell lines in the manipulation of moDC

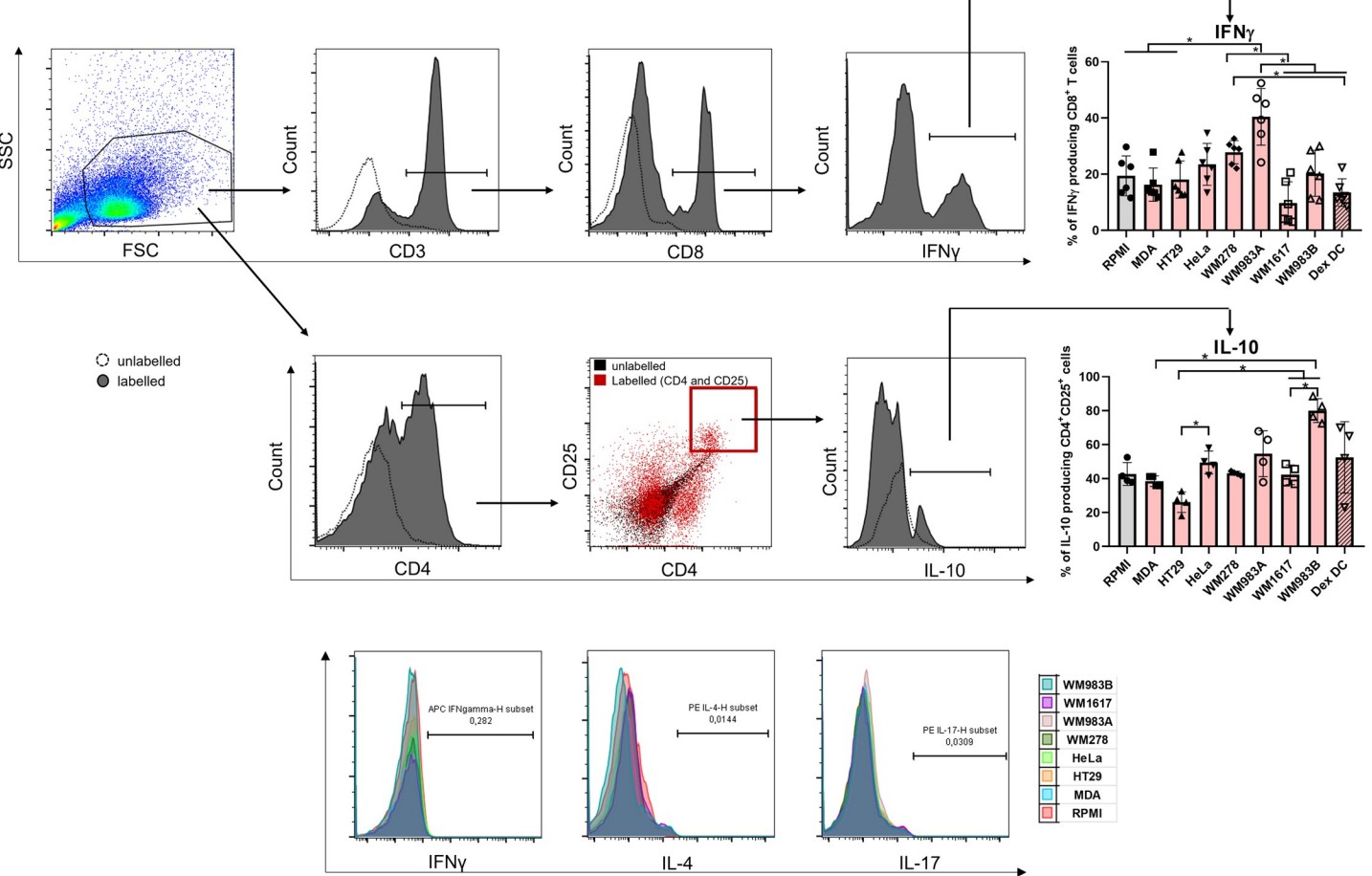

**Fig 5. Tumor cell line-educated moDCs polarize the T cells in different ways.** CD14+ monocytes were cultured with 100 ng/ml IL-4, 80 ng/ml GM-CSF ± MDA-MB231-, HeLa-, HT29-, WM278-, WM1617-, WM983A- or WM983B-CM for four days. Control or TU-CM-moDCs were co-cultured with autologous peripheral blood lymphocytes (PBL) for three, five, or nine days in RPMI-1640 medium at a ratio of 1:10 (moDCs: T-cells). The cytokine production capacity of T cells was measured by flow cytometry. The figure shows the mean plus SD of the populations positive for the measured surface molecules and cytokines calculated from at least six/ four independent experiments. Histograms or dot plots show one of the six/four independent experiments. Significance is indicated by *P < 0.05, **P < 0.01, ***P < 0.001.

differentiation,. The control moDCs (RPMI and DexDC) were not analyzed in the correlation analysis using heatmaps and correlograms (Fig 6, S6 and S7 Figs). The bioinformatic analysis disclosed that melanomas and adenocarcinomas generally induce different changes in monocyte-derived cells. However WM1617 metastatic melanoma cell line has the same effect on the moDC's function as the adenocarcinomas, but it is quite different from other melanomas (Fig 6 panels A and B). Two groups were created: moDCs differentiated in the presence of adenocarcinomas (Fig 6C, S7A Fig) or melanomas (Fig 6D, S7B Fig). The correlation analysis showed significant differences in the features of adenocarcinomas- or melanomas-edited moDCs. Stronger positive or negative correlations can be identified between the phenotypical or functional properties in the case of the moDCs differentiated in adenocarcinoma- than in the group of melanomas-conditioned moDCs. Based on our analysis, positive correlations were observed in the group of adenocarcinoma-educated monocyte-derived cells: between 1) CD14 and CD209, HLA-ABC, HLA-DR, CD86, and IL-10 production, between 2) CD209 and HLA-ABC, between 3) CD1a and CD1d, PD-L1, phagocytic activity, and TNFα/ IL-6/ IL-8-production, between 4) CD1d and TNFα/ IL-8-production, between 5) HLA-DR and CD86,

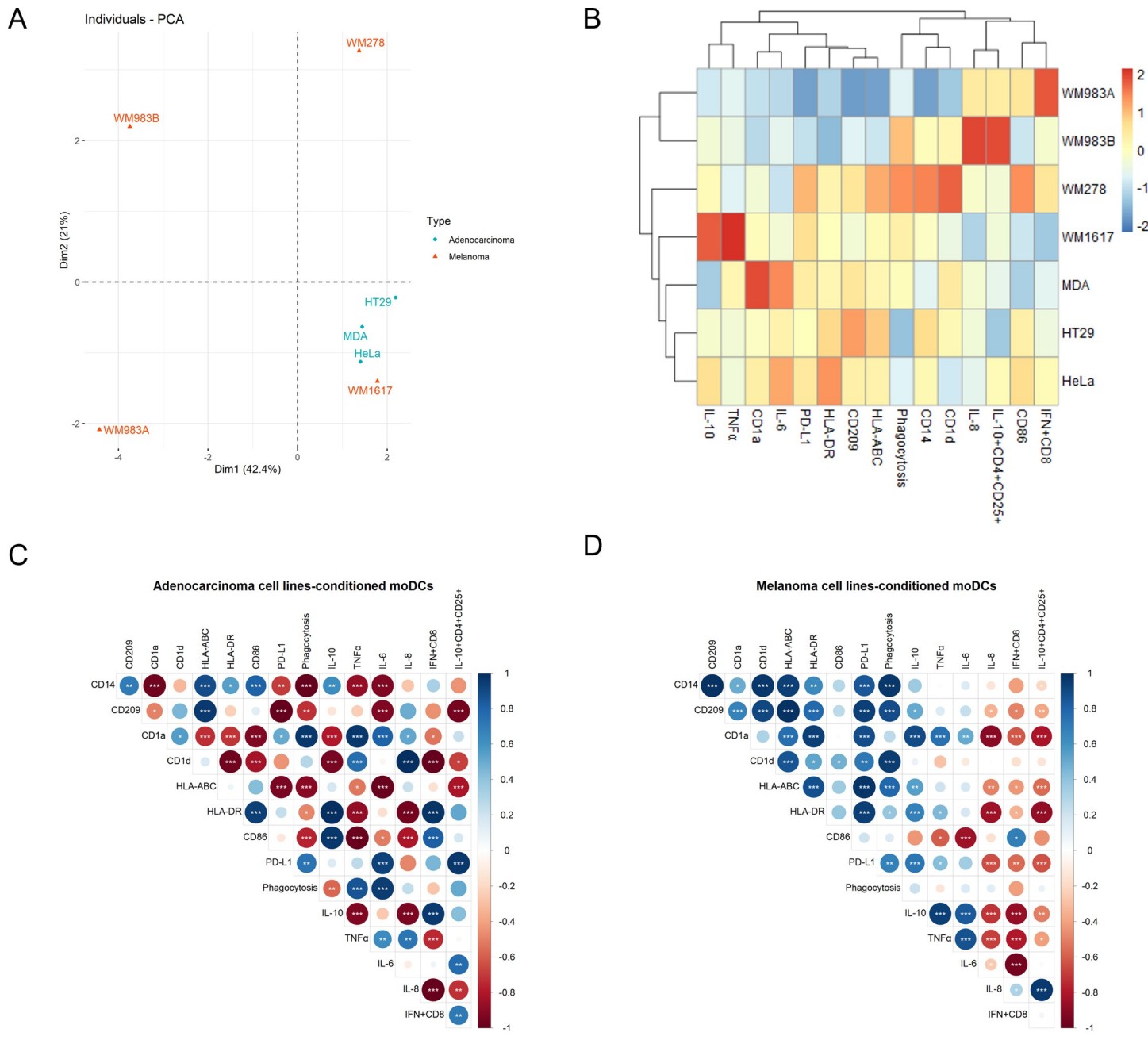

**Fig 6. Differences and similarities in phenotypical and functional properties of the different tumor-educated moDCs.** Color-coded projection of the dataset obtained from adenocarcinomas- (blue circles) and melanomas- (red triangles) conditioned moDCs on a two-dimensional PCA (A). Z-score values were calculated from the raw data, and a column-scaled heatmap (B) was generated. Panel C demonstrates the correlation between the three adenocarcinoma-educated moDCs without controls on a correlogram. The correlation between the four melanoma cell line-educated DCs is shown on Panel D. Blue circles indicate a positive correlation, while negative correlations are displayed in red. The color intensity and the size of the circle are proportional to the correlation coefficients (Pearson's correlation). Significance levels according to the P-value matrix and confidence interval matrix were defined as $^*P < 0.05$, $^{**}P < 0.01$, $^{***}P < 0.001$.

IL-10 production and IFNγ-producing CD8$^+$ T cells activatory potential, between 6) CD86 and IL-10 production and IFNγ-producing CD8$^+$ T cells activatory capacity, between 7) PD-L1 and phagocytic activity, IL-6 secretion and IL-10-producing CD4$^+$CD25$^+$ cells activatory capacity, between 8) phagocytosis and TNFα/ IL-6-production, between 9) IL-10 release

and IFNγ-producing CD8⁺ T cells activatory capacity, between 10) TNFα production and IL-6/IL-8 secretion, between 11) IL-6 and IL-10-producing CD4⁺CD25⁺ cells activatory capacity, and finally and surprisingly, between 12) the IFNγ-producing CD8⁺ T cells and IL-10-producing CD4⁺CD25⁺ cells activatory capacity (Fig 6C and S7 Fig panel A). Secondly, negative correlations were observed in the group of adenocarcinoma-educated monocyte-derived cells: between 1) CD14 and CD1a, PD-L1, phagocytic activity and TNFα/ IL-6-production, between 2) the CD209 and CD1a, PD-L1, phagocytic activity, IL-6-production and IL-10-producing CD4⁺CD25⁺ cells activatory capacity, between 3) the CD1a and HLA-ABC, HLA-DR, CD86, IL-10 secretion and IFNγ-producing CD8⁺ T cells activatory capacity, between 4) the CD1d and HLA-DR, CD86, IL-10 secretion IFNγ-producing CD8⁺ T cells and IL-10-producing CD4⁺CD25⁺ cells activatory capacity, between 5) the HLA-ABC and PD-L1, phagocytic activity, TNFα/ IL-6-production and IL-10-producing CD4⁺CD25⁺ cells activatory capacity, between 6) the HLA-DR and phagocytic activity, TNFα-production and IL-8 production, between 7) the CD86 and phagocytic activity and TNFα/IL-6/IL-8-production, between 8) the phagocytosis and IL-10 production, between 9) the IL-10 secretion and TNFα/IL-8-production, between 10) the TNFα and IFNγ-producing CD8⁺ T cells activatory capacity, finally, between the IL-8 production and T cell activator capacity (Fig 6C and S7 Fig panel A).

We continue with describing melanoma cell lines-conditioned moDCs (Fig 6D and S7 Fig panel B). Firstly, positive correlations were observed in the group of melanoma-educated monocyte-derived cells: between 1) CD14 and CD209, CD1a, CD1d, HLA-ABC, HLA-DR, PD-L1 and phagocytosis, between 2) CD209, CD1a, CD1d, HLA-ABC, HLA-DR, PD-L1, phagocytic activity and IL-10 production, between 3) CD1a and HLA-ABC, HLA-DR, PD-L1 and IL-10/TNFα/IL-6 production, between 3) CD1d and HLA-ABC, HLA-DR, CD86, PD-L1 and phagocytic activity, between 4) HLA-ABC and HLA-DR, PD-L1, phagocytic activity and IL-10 production, between 5) HLA-DR and PD-L1, phagocytic activity and IL-10/ TNFα production, between 6) CD86 and IFNγ-producing CD8⁺ T cells activatory capacity, between 7) PD-L1 and phagocytic activity and IL-10/ TNFα production, between 8) IL-10 and TNFα/IL-6 production, between 9) TNFα secretion and IL-6, finally, between 10) IL-8 production and T cell activation (Fig 6D and S7 Fig panel B). Secondly, negative correlations were observed in the group of melanoma-educated monocyte-derived cells: between 1) CD14 and IL-10-producing CD4⁺CD25⁺ cells activatory capacity, between 2) CD209, HLA-ABC, HLA-DR, and IL-8 production and IFNγ- or IL-10 producing T cell stimulatory potential, between 3) CD86 and TNFα/IL-6 production, and finally, between 4) IL-6 and IL-8 production and IFNγ-producing CD8⁺ T cells activatory capacity (Fig 6D and S7 Fig panel B). The detailed analysis of the different correlations observed between each marker in the two groups is listed in S7 Fig. In summary, PCA analysis revealed differences in the strategy of melanomas and adenocarcinoma cell lines in the manipulation of moDC differentiation, thus in their ability to orchestrate T cell responses.

## Discussion

The presence of tumor-educated tolerogenic or regulatory subsets of dendritic cells and macrophages is associated with poor prognosis in cancer patients [35, 36]. An increasing number of methods have been published to generate tumor-educated monocyte-derived cells, but almost all of these studies focus on generating tumor-associated macrophage subtypes; however, tumor-promoting DC subsets are also formed during tumorigenesis [2, 37–39]. In our study, we attempted to broaden the knowledge about adenocarcinoma- and melanoma cell line-derived mediators by comparing primary and metastatic melanoma cell lines on the differentiation of DC from monocytes.

DCs are associated with relatively high plasticity. Thus, these cells effectively adapt to the continuously changing microenvironment. According to the state of tumorigenesis, the properties and numbers of TIDCs are constantly changing [11]. Aggressive cancer growth correlates with a switch from immune-stimulatory to immune-suppressive DCs. While depletion of DCs in the early phase of the disease leads to more robust tumor expansion and aggression, in contrast, their later depletion results in the abrogation of tumor growth [40]. This suggests that cancers develop strategies to prevent DCs from responding to danger signals. Despite progress in understanding DC subsets, it remains unclear whether there are various subtypes of tolerogenic/tumor-promoting or tumor growth-preventing DC populations in the environment of different tumor types.

We investigated the effects of tumor cells on the differentiation of moDCs. During the normal process of moDCs differentiation in the presence of GM-CSF and IL-4, the expression of CD14 is down-regulated [22], while that of DC-SIGN/CD209 and CD1 family members (CD1a, b, c) [22] is increased. The reciprocal changes in the expression of CD14 and DC-SIGN are associated with the parallel loss of epigenetic markers of "activation" at the CD14 locus, but the acquisition of the same at the DC-SIGN locus [29]. Our result suggests that tumor cell lines and dexamethasone alter the epigenetic regulation of the CD14 and DC-SIGN expression in moDCs. All tumor cell type-derived factors induced the co-expression of CD14 and DC-SIGN. Based on the literature, not just the tumor cells but the tumor-associated and normal stroma cells can also modify the differentiation of moDCs in the presence of IL-4 and GM-CSF, resulting in upregulated expression of DC-SIGN [41, 42]. However, stromal cell-dependent regulation failed to decrease the monocyte marker CD14 [42]. CD1a glycolipid receptor expression on monocyte-derived cells was significantly reduced by exposure to any TU-CM or dexamethasone. In parallel, the expression of CD1d was not significantly changed by the investigated TU-CM. Based on the findings of Gerlini et al., the expression of CD1 molecules decreases on the surface of moDCs only in metastatic melanomas but not in primary melanoma [43]. In opposition to this observation, we detected a down-regulated level of CD1a molecules on moDCs upon exposure to both primary and metastatic melanoma cell lines. The decreased expression of CD1 molecules on the surface of melanoma-CM-conditioned moDCs is not surprising, as peritumoral DCs are typically CD1-negative cells as opposed to intraepidermal DCs, which are predominantly CD1a$^+$ cells. However, in sentinel lymph nodes, there is a marked appearance of mature CD1a$^+$ DCs [44]. Although different tumors uniquely modify the differentiation of monocyte into DC, we can conclude that 1. tumor-driven differentiation do not overlap with the classical tolerance (e.g., dexamethasone-induced) model, 2. all the tumors downregulated CD1a, and 3. the appearance of the CD14 and DC-SIGN double-positive population is a unique feature of the tumor-driven differentiation.

The primary function of DCs is initiating and coordinating T cell responses. These processes depend on the complex communication between DCs and T-cells via antigen-presenting and costimulatory molecules as well as through the production of cytokines. Antigen presentation of tumor-derived epitopes also requires the uptake of tumor antigens. We revealed differences in the uptake of *Lactobacillus casei* by tumor cell line-conditioned moDCs, which shows how the tumor microenvironment in some cases could influence the phagocytic capacity of DCs. Probiotic bacteria have been shown to play a role in immunomodulation and display antitumor features; the effects of various tumors on the immunomodulatory activity of microbiota could define the efficiency of checkpoint therapies [45]. As an important microbiota component, lactic acid bacteria present in the gut have been shown to have a role in the regression of carcinogenesis due to their influence on immunomodulation [46]. Smits et al. demonstrated that *L. casei* binds to DCs, and this ligation can actively prime DCs leading to the activation of IL-10 production by regulatory T cells [47]. Different tumor

cells can manipulate the uptake of probiotics and thus regulate their potential therapeutic effect. In our comparative analysis, we identified different correlations between the CD209/ DC-SIGN, CD14, and phagocytosis of *L. casei* by moDCs. These observations corroborate the importance of the simultaneous consideration of the properties of microenvironment, the phenotype and the phagocytic activity of moDCs.

We examined key cell surface molecules that contribute to antigen presentation to further study how tumor-derived supernatants can modify DC-driven T cell activation. We found no significant difference in the expression of MHCI on the surface of moDCs; however, induction of the impaired expression of MCHI is a possible escape mechanism for cancers to avoid immune responses [48]. Expressing MHCI and MHCII on antigen-presenting cells is crucial during antitumor immunity.

Based on our results, the correlations between molecules and functions in the case of adenocarcinoma cell lines-conditioned moDCs and melanoma cell lines-conditioned moDCs are pretty different. Additionally, the WM1617 metastatic melanoma cell line has the same effect on the moDC's function as the adenocarcinomas, but it is quite different from the other melanomas Our results show that cell surface-associated markers, secreted cytokines, and phagocyte activity mostly negatively correlate with the T cell cytokine production-inducing capacity in the case of the melanoma cell lines-educated moDCs'group. Our results support the hypothesis of Johnson et al., that MHC-I/II expression is required for tumor antigen presentation and may predict anti-PD-1 therapy response in the case of patients diagnosed with melanoma [49]. Interestingly, the PD-L1 expression on adenocarcinomas-conditioned moDCs positively correlated with the phagocyte activity, IL-6 production, and IL-10 producing $CD4^+CD25^+$ T cell activation. In contrast to this, in the group of moDCs altered by melanoma cell lines, the PD-L1 expression negatively correlated with the T cell response inducing capacity of monocyte-derived cells.

The cytokine production of DCs is also critical in directing the type of T cell response along with antigen presentation processes. To study DC-driven T cell polarization, we examined the autologous T cell responses after co-culturing tumor cell lines-educated moDCs and peripheral blood lymphocytes.

Based on the summary of literature data, tumor growth facilitates the induction and recruitment of $CD4^+$ regulatory T cells that secrete IL-10 and TGF-β [50] and suppress effector $CD8^+$ T cell responses [51]. Overall, TU-CM-educated DCs induced Treg differentiation, especially the subtype of IL-10 producing Tregs. In our system, only DCs treated with the conditioned media of WM1617 and WM983B metastatic melanoma cell lines decreased the secretion of IFNγ by $CD8^+$ T cells compared to its primary counterpart. However, WM983B-moDCs stimulated the production of IL-10 by $CD4^+CD25^+$T cells most intensively compared to the other WM1617 cell line. The observation can explain this that IFNγ acts as an essential factor that impairs IL-10 immunosuppressive and anti-inflammatory activity [52]. The IL-10 has been published to be expressed by various melanoma cells, especially metastatic malignancy [53]. However, melanoma cell lines did not produce IL-10 cytokine in our experiments. Thus it seems to be a DC-dependent process.

TNFα and IL-6 were initially considered pro-inflammatory molecules, but current preclinical and clinical data have shown that TNFα also mediates a paradoxical anti-inflammatory and immunomodulatory effect [54]. Pierini et al. demonstrated that peripheral blood of recipient animals during acute GVHD induced $CD4^+CD25^+FoxP3^+$ Treg activation and enhanced their function. They found that the phenomenon could be explained by the priming of these cells by the myeloid cells-derived TNF-α, which selectively activates Tregs without impacting $CD4^+FoxP3^-$ T cells [55]. We detected upregulated production of TNFα and IL-10 by WM1617 metastatic melanoma cell-line-educated-moDCs. In the group of melanoma-CM-

educated moDCs, TNFα secretion negatively correlated with the ability of moDCs to induce IL-10 producing CD4$^+$CD25$^+$ T cells. IL-6 is critical for Th17 cell induction [56] and unfavorable for Treg induction. However, we found that adenocarcinomas-educated moDCs' IL-6 production positively correlates with IL-10 producing T cell activator capacity by monocyte-derived cells. IL-8 is a chemokine produced by, e.g., cancer or cancer-associated cells and whose serum concentration correlates with poor prognosis in the case of cancer patients. Every tumor cell line-conditioned moDC produced IL-8. Besides, IL-8 was directly secreted and by MDA and melanoma cell lines, except for WM1617. This chemokine attracts myeloid-derived suppressor cells (MDSC) and activates granulocytic MDSC to extrude DNA-forming neutrophil extracellular traps (NET) [57]. These mechanisms and IL-8 production of tumor cell-lines-educated moDCs are likely relevant in shaping a pro-tumoral leukocyte microenvironment in cancer. Based on our analysis, the remarkable difference can be identified when we observe the IL-8 production and the effect of moDCs on T cell activation; while in the case of IL-8 producing adenocarcinomas-altered moDC's T cell activator capacity is impaired, the melanoma cell lines-conditioned moDCs IL-8 secretion was associated with their T cell response evoking ability.

Taken together, our project could contribute to broadening knowledge about the effect of different tumor cell lines on the differentiation of moDCs. Until now, more study was available on the investigation of TAMs and TAM-differentiation protocols using tumor cell-derived conditioned media. We firstly provide a comparative analysis of the tumor-edited-moDCs differentiated in the presence of adenocarcinomas- or melanomas (primary/metastatic)- derived soluble mediators. We analyzed our observations using bioinformatics tools and found differences between the phenotypic and functional properties of moDCs formed with CM of adenocarcinomas or melanomas. This way, we could offer new *in vitro* differentiation protocols to study various tumor-conditioned moDCs. Our results highlight the differences in the effects of different cancer cell lines in the regulation of moDCs, as adenocarcinomas, regardless of the tissues of origin, manipulated DCs differently than melanomas independently of their metastatic activity.

## Supporting information

**S1 Fig. To determine which T-lymphocyte populations are responsible for the cytokine production, the T cells were stimulated with 1 μg/ml ionomycin and 20 ng/ml phorbol-myristic acetate (PMA) for 4 hours.** The vesicular transport was inhibited by BD GolgiStop™ protein transport inhibitor (BD Biosciences) after the activation. The cells were then labeled with anti-human CD3-FITC, and anti-human CD8-PE or CD4-PerCP and anti-human CD25-PE antibodies. The samples were then fixed and permeabilized and labeled with ananti-human CD8-PE or CD4-PerCP and with anti-human IFNγ-APC, anti-human IL-4-PE, anti-human IL-10-Alexa Fluor 488, anti-human IL-17-Pe antibodies. Fluorescence intensities were measured by Novocyte2000R Flow Cytometer.
(TIF)

**S2 Fig. Effect of TU-CMs on the viability of moDCs.** Cell viability was assessed by 7-aminoactinomycin-D (7-AAD; 10 μg/ml) staining. Samples were stained for 15 minutes with 7-AAD immediately before flow cytometric analysis. Fluorescence intensities were measured by Novocyte2000R Flow Cytometer, and data were analyzed by the FlowJo v X.0.7 software.
(TIF)

**S3 Fig. TU-CMs did not change the autofluorescence of moDCs.** CD14+ monocytes were cultured with 100 ng/ml IL-4, 80 ng/ml GM-CSF ± MDA-MB231-, HeLa-, HT29-, WM278-,

WM1617-, WM983A-, WM983B-CM or in the presence of dexamethasone for four days. On day 4, the autofluorescence of moDCs was measured by flow cytometry. Histograms show one representative experiment.
(TIF)

**S4 Fig. TU-CMs modify the phenotype of dendritic cells.** CD14+ monocytes were cultured with 100 ng/ml IL-4, 80 ng/ml GM-CSF ± MDA-MB231-, HeLa-, HT29-, WM278-, WM1617-, WM983A- or WM983B-CM for four days. On day 4, the expression of CD163 and CD206 were analyzed by flow cytometry. Figures show the mean plus SD values of the populations positive for the measured surface molecules calculated from three experiments. Histograms show one of the three independent experiments. Significance is indicated by $^*$P < 0.05.
(TIF)

**S5 Fig. TU-CMS alters the cell surface expression of molecules involved in conventional T cell activation.** CD14+ monocytes were cultured with 100 ng/ml IL-4, 80 ng/ml GM-CSF ± MDA-MB231-, HeLa-, HT29-, WM278-, WM1617-, WM983A- or WM983B-CM for four days. On day 4, the cell surface expression of HLA-ABC, HLA-DR, CD86, and PD-L1 was analyzed by flow cytometry. The figure shows the MFI plus SD calculated from five independent experiments. Significance is indicated by $^*$P < 0.05.
(TIF)

**S6 Fig. Differences and similarities in phenotypical and functional properties of the different tumor-educated and control moDCs.** Color-coded projection of the dataset obtained from adenocarcinomas- (blue circles) and melanomas- (red squares) conditioned and control (grey triangles) moDCs on a two-dimensional PCA (A). Z-score values were calculated from the raw data, and a column-scaled heatmap (B) was generated.
(TIF)

**S7 Fig. Differences and similarities in phenotypical and functional properties of the different tumor-educated moDCs.** Panel A demonstrates a mixed correlogram from the data of the three adenocarcinoma-educated moDCs without controls. On the lower part, the individual correlation matrix values are shown. Blue circles indicate positive correlation on the upper part, while negative correlations are displayed in red. The color intensity and the size of the circle are proportional to the correlation coefficients (Pearson's correlation). The correlation between the four melanoma cell line-educated DCs is shown in Panel B. On the lower part, the individual correlation matrix values are shown. Blue circles indicate positive correlation on the upper part, while negative correlations are displayed in red. The color intensity and the size of the circle are proportional to the correlation coefficients (Pearson's correlation).
(TIF)

## Acknowledgments

We thank Erzsike Nagyné Kovács for her excellent technical assistance.

## Author Contributions

**Conceptualization:** Ramóna Kovács, Márta Tóth, Gábor Koncz, Anett Mázló.

**Data curation:** Ramóna Kovács, Márta Tóth, Tímea Szendi-Szatmári, Anett Mázló.

**Formal analysis:** Márta Tóth, Anett Mázló.

**Funding acquisition:** Attila Bácsi, Gábor Koncz.

**Investigation:** Ramóna Kovács, Gábor Koncz, Anett Mázló.

**Methodology:** Sára Burai, Ramóna Kovács, Tamás Molnár, Márta Tóth, Tímea Szendi-Szatmári, Viktória Jenei, Zsuzsanna Bíró-Debreceni, Shlomie Brisco, Gábor Koncz, Anett Mázló.

**Resources:** Margit Balázs, Attila Bácsi, Gábor Koncz.

**Software:** Márta Tóth.

**Supervision:** Attila Bácsi, Gábor Koncz, Anett Mázló.

**Validation:** Ramóna Kovács, Tamás Molnár, Márta Tóth, Zsuzsanna Bíró-Debreceni, Anett Mázló.

**Visualization:** Sára Burai, Márta Tóth, Anett Mázló.

**Writing – original draft:** Márta Tóth, Viktória Jenei, Shlomie Brisco, Gábor Koncz, Anett Mázló.

**Writing – review & editing:** Ramóna Kovács, Márta Tóth, Margit Balázs, Attila Bácsi, Gábor Koncz, Anett Mázló.

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
