## [Decision Letter · Decision Letter 0]

29 Mar 2022

PONE-D-21-38603Comprehensive analysis of different tumor cell-line produced soluble mediators on the differentiation and functional properties of monocyte-derived dendritic cellsPLOS ONE

Dear Dr. Türk-Mázló,

Thank you for submitting your manuscript to PLOS ONE. After careful consideration, we feel that it has merit but does not fully meet PLOS ONE’s publication criteria as it currently stands. Therefore, we invite you to submit a revised version of the manuscript that addresses the points raised during the review process.

We look forward to receiving your revised manuscript.

Kind regards,

Jianhong Zhou

Associate Editor

PLOS ONE

Journal Requirements:

2. Please update your submission to use the PLOS LaTeX template. The template and more information on our requirements for LaTeX submissions can be found at http://journals.plos.org/plosone/s/latex

3. We note that you have included the phrase “data not shown” in your manuscript. Unfortunately, this does not meet our data sharing requirements. PLOS does not permit references to inaccessible data. We require that authors provide all relevant data within the paper, Supporting Information files, or in an acceptable, public repository. Please add a citation to support this phrase or upload the data that corresponds with these findings to a stable repository (such as Figshare or Dryad) and provide and URLs, DOIs, or accession numbers that may be used to access these data. Or, if the data are not a core part of the research being presented in your study, we ask that you remove the phrase that refers to these data

Reviewers' comments:

Reviewer's Responses to Questions

**Comments to the Author**

1. Is the manuscript technically sound, and do the data support the conclusions?

Reviewer #1: Yes

Reviewer #2: Partly

2. Has the statistical analysis been performed appropriately and rigorously? 

Reviewer #1: Yes

Reviewer #2: I Don't Know

3. Have the authors made all data underlying the findings in their manuscript fully available?

Reviewer #1: No

Reviewer #2: Yes

4. Is the manuscript presented in an intelligible fashion and written in standard English?

Reviewer #1: Yes

Reviewer #2: Yes

5. Review Comments to the Author

Reviewer #1: Referee comments on PONE-D-21-38603

Burai, S, et al “Comprehensive analysis of different tumor cell-line produced soluble mediators on the differentiation and functional properties of monocyte-derived dendritic cells”

Türk-Mázló and colleagues investigate the impact of tumor environment on dendritic cell differentiation. To do so, they analyze and compare the effect of the tumor cell line-derived condition media using eight different adenocarcinoma and melanoma cell lines on in vitro differentiation of dendritic cells according to several variables. Their results highlight the different effects on DC differentiation between melanoma and adenocarcinoma cell lines.

Overall, they provide an interesting approach for the in vitro study of different tumor-promoting DCs, however, revisions seem to be needed.

Major revisions:

1) There are several inconsistencies between the text and the results presented in the figures. In particular:

“HT29-, HeLa-CM-moDCs and in accordance with the literature, DexDCs also showed significantly higher CD14 expression [23] compared to the control moDCs (Fig 1A).” (lines 270-271): Fig 1 does not show significant result for HeLa cell line.

“HT29, WM278, WM1617, and WM983B-conditioned moDCs expressed significantly more CD14 and CD209 molecules than their MDA- or WM983A-educated counterparts.” (lines 280-282): This should be re-worded to refer to higher frequency of co-expression of CD14 and CD209 instead of more expression. Fig 1 also does not show that the difference between WM1617 and MDA or WM983A cell lines is significant.

“… significant difference was observed between the WM983A and DexDC groups).” (line 293): this difference is between DexDC and WM278 and not between DexDC and WM983A.

“… the phagocytic capacity was increased slightly but not significantly compared to control moDCs. These results indicate that only certain tumors can alter the phagocytic capacity of the DCs.” (line 374-376) and “… indicating that tumor microenvironment influences the phagocytic capacity of DCs” (line 522): The authors state that the results indicate that only certain tumors may impair or influence the phagocytic capacity of DCs, stating that no significant results are shown. I think the authors should be less categorical in the Discussion.

Fig 5 needs more thorough description of gating. Panel A would be better visualized as a 2D plot for CD4 and CD8 instead of independent density/histogram plots. For the quantification on the right it is not entirely clear what the denominators are. CD8+ IFNg+ should probably be reported as a percentage of total CD8+ cells and similarly CD4+CD25+FOXP3+ should be percentage of total CD4+. It is not clear whether CD4+CD25+IL10+ should be reported as percentage of CD4+ or CD4+CD25+.

“Based on our correlation analysis, between HLA-ABC, HLA-DR and PD-L1, their expression was similarly regulated in the case of melanoma cell lines-educated DCs, whereas strong negative correlation was observed in the group of moDCs conditioned by adenocarcinomas with the respect of HLA-ABC, HLA-DR and PD-L1 expressions.” (lines 539-543): Regarding adenocarcinoma cell lines, only HLA-ABC and PD-L1 are strongly anti-correlated, the other pairs (HLA-ABC with HLA-DR and HLA-DR with PD-L1) are just not correlated. This should be described more accurately.

2) There are some errors in the annotation of the figures and the accompanying descriptions. I encourage the authors to review the annotation of their figures and their descriptions. In particular:

Lines 291 and 301 refer to Fig 1C, however, Figure 1 contains only two parts (Fig 1A and Fig 1B).

Presenting Fig 3B as a heatmap makes it very difficult to compare with Fig 3A. Can another visualization be found (e.g. perhaps the same as Fig 3A) that makes these results easier to interpret?

The references to Fig 5 do not target the correct part of the figure. For example, the reference to Fig 5C on line 408 should probably be in the previous sentence and replaced here by Fig 5D.

There is an annotation problem between Fig 6 and its description (the description mentions Correlogram B and C while the correlograms make up Fig 6C and Fig6D).

3) The authors propose an interesting approach using bioinformatics to enrich the interpretation of their results. Although the authors have the good idea to use the correlation circle as a complement to the PCA, they did not apply it on the variables (as described in the figure description) but on the cell lines. Correcting this error and interpreting the results obtained could bring possibly interesting information and help to enrich the bioinformatic results.

“In summary, PCA analysis revealed differences in the strategy of melanomas and adenocarcinoma cell lines in the manipulation of moDC differentiation, thus in their ability to orchestrate T cell responses.” (lines 449-451): PCA shows clear differences between the cell lines and the control (RPMI and DexDC) but it doesn’t show differences between the different cell lines. Can the authors perhaps provide a PCA after removing the RMPI and DexDC controls to see if it validates this conclusion?

Minor revisions:

1) A better description of the method of the bioinformatics part would be helpful (package version, function parameters, ...).

2) Adding titles to the figures would help the reader to follow (e.g.: add adenocarcinoma above the corresponding correlogram, ...)

3) Improve the title of figure 6 "PCA" which does not take into account the half of the figure.

4) Figure2: results observe using MFI histograms and the barplots are sometimes different, perhaps more representative MFI histograms should be used instead (e.g.: HLA-ABC of WM983B cell line shows clear differences by MFI plot at top of Fig 2A but much more moderate typical effect in barplot below).

5) “Our result suggests that tumor cell lines and dexamethasone alter the epigenetic regulation of the CD14 and DC-SIGN expression in moDCs. All tumor cell type-derived factors induced the co-expression of CD14 and DC-SIGN.” (lines 495-498): Authors say cell lines and dexamethasone alter the epigenetic regulation of CD14 and DC-SIGN expression in moDCs, however, only CD14 expression is altered as described after, maybe this sentence could be reformulated to explain better this observation.

6) “We have found that the expression pattern of MHCII and PD-L1 was also correlated with the ability of TU-CM-conditioned moDCs to stimulate CD4+CD25+FoxP3+ T cells. (line 545-547) : Authors could be precise these observation is in the melanoma cell lines.

7) “Overall, TU-CM-educated DCs induced Treg differentiation, especially the subtype of IL-10 producing Tregs.” (lines 554-555): Add the figure reference.

8) “… Interestingly, this correlation was the opposite in the case of melanoma cell-line-CM-conditioned moDCs” (lines 576-577): Authors precise the TNFα and CD4+CD25+FoxP3+ T cells positively correlate in melanoma cell lines, however, only WM1617 cell line expresses TNFα well. Authors should describe this observation more carefully.

9) “IL-8 production by adenocarcinomas negatively correlated with the expression of MHC and costimulatory molecule CD86 as well as IL-10 producing helper T cell polarizing capacity of moDCs” (lines 585-587): It is only the case with MHCII, not MHCI.

Suggestions:

1) Unless I have not seen the file, it would be useful to provide an excel or csv file containing all the variables used for the bioinformatics analyses.

2) For the correlation analyses, adding in additional figures the plots of correlations of the most interesting pairs of variables could help to better understand the data. Moreover, it could allow to see if some melanoma or adenocarcinoma cell lines behave differently from others.

3) To complete PCA, authors could be produce a a clustering/heatmap on the same data.

Reviewer #2: In this manuscript, the authors aim to decipher the impact of tumor cell-derived soluble factors on monocyte differentiation into monocyte-derived DC (mo-DC). They use a well described in vitro model (GM-CSF + IL-4). The authors claim that monocyte exposure to factors derived from tumor cell modify the phenotype of mo-DC and their expression of co-stimulatory molecules. The authors next focus on typical functions of mo-DC: phagocytosis and polarization of T cells. They conclude that tumor-derived soluble factors alter phagocytosis and polarization of T cells. In the last part, the authors show correlations between different properties or functions of mo-DC: phenotype, cytokine secretion, ability to polarize T cells and phagocytosis.

Major comments:

1. In figure 1, the authors define mo-DC as CD14+ or CD209+. However, CD14 and CD209 are not specific markers of mo-DC and can be expressed by macrophages too. Moreover, it has been shown by mass cytometry that the mo-DC population (differentiated from monocytes with GMCSF and IL-4) is homogeneous, with some cells expressing CD163 and MERTK suggesting the presence of mo-mac in this model (Sander, Immunity 2017). Are the authors sure that tumor cell derived factors modify only surface markers and not the identity of the cells? In line with this comment, the authors show that CD1a (typical mo-DC marker) expression is decreased by factors secreted by tumor cells.

2. Figure 2 : it is not clear how the authors define the positive population for the different markers. In comparison with the unlabeled control, it seems that almost all the cells are positive. They should rather analyze the MFI.

3. Figure 3 : the authors show that some tumor cell lines secrete IL8 and IL6. When measuring the secretion of cytokines by mo-DC, did the authors use a control to exclude cytokines coming from the conditioned medium?

4. In figures 4 and 5, the authors should increase the n. It is hard to reconcile how they obtain statistically significant results with n=3.

5. In figure 6, the staining for Foxp3 should be improved. A better staining allowing the identification of T regs could modify the conclusion.

Minor comments:

1. In the introdcution, the authors should mention the DC1 population. It has been shown that this subset of DC is involved in the anti-tumor responses.

2. In the section « DC-T cell co-cultures » of the material and methods, more details about the procedure are needed. How were T cells isolated? Were CD8 and CD4 T cells cultured together with mo-DC (3 cell types in the same well)? Did the authors use frozen PBMCs to isolate T cells (as they fo autologous co-cultures)?

3. The authors should specify which statistical tests were performed.

4. In figure 1 and 2, the authors should include an FMO for each condition and not only the control one (RPMI).

5. The authors should explain why they choose to investigate the phagocytosis of bacteria in the context of anti-tumor immunity.

6. In figure 5, when gating T cells, the authors should not exclude the cells with higher FSC and SSC. After proliferation, T cells become bigger and more granular.

7. In figure 5D, it is not clear why the authors gate on two different populations (black and red).

8. I would recommend the authors to improve the language to facilitate the readability of the text.

6. PLOS authors have the option to publish the peer review history of their article (what does this mean?). If published, this will include your full peer review and any attached files.

Reviewer #1: No

Reviewer #2: No

---

## [Author Response · Author response to Decision Letter 0]

26 May 2022

Thank you very much for the review of our manuscript entitled “Comprehensive analysis of different tumor cell-line produced soluble mediators on the differentiation and functional properties of monocyte-derived dendritic cells ” and the opportunity to resubmit a revised version. 

We greatly appreciate the positive evaluation of our manuscript and the valuable comments of the Reviewers. We addressed the comments of the reviewer. To enhance the quality of our manuscript we have conducted additional experiments and made several changes in the text. 

As the Respond to Reviewers exceeds 20,000 characters, it is uploaded as separated file. 

Sincerely,

Anett Mázló

---

## [Decision Letter · Decision Letter 1]

24 Jun 2022

PONE-D-21-38603R1Comprehensive analysis of different tumor cell-line produced soluble mediators on the differentiation and functional properties of monocyte-derived dendritic cellsPLOS ONE

Dear Dr. Mázló,

Thank you for submitting your manuscript to PLOS ONE. After careful consideration, we feel that it has merit but does not fully meet PLOS ONE’s publication criteria as it currently stands. Therefore, we invite you to submit a revised version of the manuscript that addresses the points raised during the review process.

We look forward to receiving your revised manuscript.

Kind regards,

Jean Kanellopoulos, M.D., Ph.D.

Academic Editor

PLOS ONE

Additional Editor Comments:

Dear Dr Anett Mázló,

You will find the comments of reviewers N°1 and N°2. They contain some requests and criticisms that need to be addressed carefully.

I would like to stress that the flow cytometry analyses must be clarified as requested by reviewer N°2. For figure N°4, a negative control must be introduced in typical experiments.

For figure N°2, you must precise how you define the positive populations and modify your initial conclusions if needed.

Yours Sincerely,

Dr Jean Kanellopoulos

Reviewers' comments:

Reviewer's Responses to Questions

**Comments to the Author**

1. If the authors have adequately addressed your comments raised in a previous round of review and you feel that this manuscript is now acceptable for publication, you may indicate that here to bypass the “Comments to the Author” section, enter your conflict of interest statement in the “Confidential to Editor” section, and submit your "Accept" recommendation.

Reviewer #1: All comments have been addressed

Reviewer #2: (No Response)

2. Is the manuscript technically sound, and do the data support the conclusions?

Reviewer #1: Yes

Reviewer #2: Partly

3. Has the statistical analysis been performed appropriately and rigorously? 

Reviewer #1: Yes

Reviewer #2: Yes

4. Have the authors made all data underlying the findings in their manuscript fully available?

Reviewer #1: Yes

Reviewer #2: Yes

5. Is the manuscript presented in an intelligible fashion and written in standard English?

Reviewer #1: Yes

Reviewer #2: Yes

6. Review Comments to the Author

Reviewer #1: Referee comments on PONE-D-21-38603

Burai, S, et al “Comprehensive analysis of different tumor cell-line produced soluble mediators on the differentiation and functional properties of monocyte-derived dendritic cells”

The revised version of the manuscript by Mázló and colleagues is significantly improved and addresses most of the original comments. I now recommend accepting the manuscript for publication although there are some essential, minor revisions that must be addressed (below). I trust the editor to evaluate these revisions, I do not need to see the manuscript again.

Minor revisions:

1) References should be added for statements on lines 100-107 and lines 627-628.

2) The description of the methods concerning the bioinformatics part is really much better. However, it is still necessary to add the parameter values used for these commands, even if just the default values were used. Similarly, the parameter values for pheatmap in figure 6 and S6 should be specified.

3) The text describing the results in Figure 3 is unclear.

The statement quoted in line 371-372 should reference Figure 3B instead of 3A. Conversely, the statement on line 381-382 should refer to Figure 3A.

Similarly, the statement in line 378-381 concerning WM1617 should refer to Figure 3A instead of 3B.

4) Figure 4B legend should be revised to refer to boxplots instead of histograms.

5) Figures 5 and 6 appear to have been swapped or mislabeled. Eg lines 422- 451 appear to refer to the figure labeled 6 instead of 5 and lines 453-530 refer to the figure labeled 5 instead of 6.

6) The two sentences on line 460-463 should be joined and reworded for clarity.

7) Several errors are present in the description of the correlation analyses of adenocarcinomas: CD1a and HLA-ABC are not positively correlated (line 475), CD1a is not negatively correlated with IL10-producing CD4+CD25+ but is negatively correlated with IFNγ-producing CD8+ cells (line 489) and HLA-DR is not negatively correlated with IFNγ-producing CD8+ but with IL8-production (line 493-494). In addition, t might be better to specify the correlations grouped in the term “T cell stimulatory potential” (line 513), such as: “…T cell stimulatory potential (IFNγ-producing and T cell activator)”.

8) There are some small errors in the discussion:

There is an error in line 622, WM278 should be replaced by WM1617.

There is an error in the sentence of line 638-639: the negative correlation between TNFα secretion and IL-10 producing CD4+CD25+ T cells is observed in melanoma and not in adenocarcinoma.

9) Several typos are present in the manuscript such as: "DC-SIGN/CD20916" (line 553) instead of "DC-SIGN/CD209", "DCsand" (line 589) instead of "DCs and", etc. Extensive proof reading is required.

Reviewer #2: The flow cytometry data still needs to be improved.

In figure 2 : it is still not clear how the authors define the positive population for the different phenopytpic markers. If the positive population is determined according to the unlabelled cells, the mean should be higher than 80% (and not 50% as shown in the graphs).

In figure s4 : a representative facs plot should be shown for CD163 and CD206.

In figure 4 : a negative control (monocyte-derived cells + Bacteria at 4°C) is lacking. The fluorescence measured should correspond to binding of the cell surface instead of phagocytosis.

7. PLOS authors have the option to publish the peer review history of their article (what does this mean?). If published, this will include your full peer review and any attached files.

Reviewer #1: No

Reviewer #2: No

---

## [Author Response · Author response to Decision Letter 1]

8 Aug 2022

Response to Reviewers

Thank you very much for the review of our manuscript entitled “Comprehensive analysis of different tumor cell-line produced soluble mediators on the differentiation and functional properties of monocyte-derived dendritic cells” and the opportunity to resubmit a revised version. 

We greatly appreciate the positive evaluation of our manuscript and the valuable comments of the Reviewers and the Editor. We addressed all the comments of the Reviewers. To enhance the quality of our manuscript we have conducted additional experiments and made several changes in the text. Please see our detailed responses below.

Reviewer#1:

1) References should be added for statements on lines 100-107 and lines 627-628.

First of all, we are very grateful to the reviewer for a thorough review of our manuscript. Additionally, we thank you for bringing errors to our attention. We apologize for the inconsistencies between the text and the results presented in the figures. We corrected the highlighted sentences, refined the statements and conclusions. We checked the consistency of the text and figures as well as the text were edited in the revised manuscript. In the revised version of the manuscript, the text was aligned with the figures and panels in all cases.

We completed the text with the next references:

line 108: 

Krempski J, Karyampudi L, Behrens MD, Erskine CL, Hartmann L, Dong H, Goode EL, Kalli KR, Knutson KL. Tumor-infiltrating programmed death receptor-1+ dendritic cells mediate immune suppression in ovarian cancer. J. Immunol. 2011;186:6905–6913.

line 635:

Itakura E, Huang RR, Wen DR, Paul E, Wunsch PH, Cochran AJ: IL-10 expression by primary tumor cells correlates with melanoma progression from radial to vertical growth phase and development of metastatic competence. Mod Pathol. 2011, 24: 801-809. 10.1038/modpathol.2011.5.

2) The description of the methods concerning the bioinformatics part is really much better. However, it is still necessary to add the parameter values used for these commands, even if just the default values were used. Similarly, the parameter values for pheatmap in figure 6 and S6 should be specified.

As requested by the reviewer, we have completed the M&M with the specified parameter values: lines 256-272

Bioinformatical Correlation Analyses

R (version 4.1.3) [16] and RStudio (version 1.4.1717) were used for bioinformatical analyses. Dataset contained the mean of marker expression, cytokine secretion, T-cell polarizing and phagocytosis data. Correlation matrices were performed by R base package (v 4.1.3) with cor function for adenocarcinoma and melanoma cell lines-conditioned moDCs separately. Correlograms from correlation matrices were plotted by corrplot (v 0.92) package [17] with corrplot function. P-value matrix and confidence intervals matrix were added using the cor.mtest function of corrplot package. Significance levels were defined as *P < 0.05; **P < 0.01; ***P < 0.001. Mixed correlograms were performed by corrplot.mixed function of the corrplot package. Heatmap was generated from the raw data by pheatmap function of pheatmap (v 1.0.12) library [18]. Principal Component Analysis (PCA) was made by PCA function of FactoMiner package (v 2.4) [19] from raw data with or without controls (RPMI and dexDC) separately. Results of the PCA were visualized by factoextra package (v 1.0.7) [20] with fviz_pca_ind function. 

is completed as it is highlighted below:

Bioinformatical Analyses

R (version 4.1.3) [16] and RStudio (version 1.4.1717) were used for bioinformatical analyses. Dataset contained the mean of marker expression, cytokine secretion, T-cell polarizing and phagocytosis data. Correlation matrices were performed by R base package (v 4.1.3) with cor function for adenocarcinoma and melanoma cell lines-conditioned moDCs separately. Correlograms from correlation matrices were plotted by corrplot (v 0.92) package [17] with corrplot function using the following parameters: type = upper, order = original. P-value matrix and confidence intervals matrix were added using the cor.mtest function with confidence level 0.95 of corrplot package. Significance levels were defined as *P < 0.05; **P < 0.01; ***P < 0.001. Mixed correlograms were performed by corrplot.mixed function of the corrplot package with the parameters lower = number, upper = circle, order = original. Heatmap was generated from the raw data by pheatmap function of pheatmap (v 1.0.12) library [18] using the scale = column parameter. Principal Component Analysis (PCA) was made by PCA function of FactoMiner package (v 2.4) [19] using the default values of scale.unit, ncp and graph parameters from raw data with or without controls (RPMI and dexDC) separately. Results of the PCA were visualized by factoextra package (v 1.0.7) [20] with fviz_pca_ind function with default parameters. 

3) The text describing the results in Figure 3 is unclear.

We apologize for the inconsistencies between the text and the results presented in the figures.

The statement quoted in line 371-372 should reference Figure 3B instead of 3A. Conversely, the statement on line 381-382 should refer to Figure 3A.

As it is labelled in the revised manuscript, we corrected the highlighted sentence.

Similarly, the statement in line 378-381 concerning WM1617 should refer to Figure 3A instead of 3B.

As it is labelled in the revised manuscript, we corrected the highlighted sentence.

4) Figure 4B legend should be revised to refer to boxplots instead of histograms.

Mentioned sentence is corrected in the revised manuscript:

“Diagram shows the mean plus SD of seven independent experiments. Boxplot shows one of the seven independent experiments.”

5) Figures 5 and 6 appear to have been swapped or mislabeled. Eg lines 422- 451 appear to refer to the figure labeled 6 instead of 5 and lines 453-530 refer to the figure labeled 5 instead of 6.

In this case we did not find mistakes, however, in the automatically generated pdf file the sequence of Figure 5 and 6 was different, but the numbers of the given figures were correct. In this regard, we will carefully check the merged uploaded files.

6) The two sentences on line 460-463 should be joined and reworded for clarity.

We apologize for the incorrect wording. Mentioned sentence is corrected in the revised manuscript:

“To reveale differences in the strategy of melanomas and adenocarcinoma cell lines in the manipulation of moDC differentiation.”

is changed to:

“To reveal differences in the strategy of melanomas and adenocarcinoma cell lines in the manipulation of moDC differentiation bioinformatic analysis was performed.”

7) Several errors are present in the description of the correlation analyses of adenocarcinomas: CD1a and HLA-ABC are not positively correlated (line 475), CD1a is not negatively correlated with IL10-producing CD4+CD25+ but is negatively correlated with IFNγ-producing CD8+ cells (line 489) and HLA-DR is not negatively correlated with IFNγ-producing CD8+ but with IL8-production (line 493-494). In addition, t might be better to specify the correlations grouped in the term “T cell stimulatory potential” (line 513), such as: “…T cell stimulatory potential (IFNγ-producing and T cell activator)”.

We apologize for the inconsistencies between the text and the results presented in the figures.

In the corrected manuscript the next changes are labelled:

 “CD1a and CD1d, HLA-ABC, PD-L1, phagocytic activity, and TNFα/ IL-6/ IL-8-production”

is changed to:

“CD1a and CD1d, PD-L1, phagocytic activity, and TNFα/ IL-6/ IL-8-production”

Additionally:

“the CD1a and HLA-ABC, HLA-DR, CD86, IL-10 secretion and IL-10-producing CD4+CD25+ cells activatory capacity”

is changed to:

“the CD1a and HLA-ABC, HLA-DR, CD86, IL-10 secretion and IFNγ-producing CD8+ T cells activatory capacity”

Additionally:

“the HLA-DR and phagocytic activity, TNFα-production and IFNγ-producing CD8+ T cells activatory capacity”

is changed to:

“the HLA-DR and phagocytic activity, TNFα-production and IL-8 production”

Additionally:

“CD209, CD1a, HLA-ABC, HLA-DR, PD-L1, IL-10 or TNFα and IL-8 production and T cell stimulatory potential”

is changed to: 

“CD209, HLA-ABC, HLA-DR and IL-8 production and IFNγ- or IL-10 producing T cell stimulatory potential”

8) There are some small errors in the discussion:

There is an error in line 622, WM278 should be replaced by WM1617.

The mistake is corrected in the revised manuscript.

“In our system, only DCs treated with the conditioned media of WM278 and WM983B metastatic melanoma cell lines decreased the secretion of IFNγ by CD8+ T cells compared to its primary counterpart.”

is changed to:

“In our system, only DCs treated with the conditioned media of WM1617 and WM983B metastatic melanoma cell lines decreased the secretion of IFNγ by CD8+ T cells compared to its primary counterpart.”

There is an error in the sentence of line 638-639: the negative correlation between TNFα secretion and IL-10 producing CD4+CD25+ T cells is observed in melanoma and not in adenocarcinoma.

The mistake is corrected in the revised manuscript.

“In oppose to the moDCs conditioned by melanomas, in the group of adenocarcinoma-CM-educated moDCs TNFα secretion negatively correlated with the ability of moDCs to induce IL-10 producing CD4+CD25+ T cells.”

is changed to:

“In the group of melanoma-CM-educated moDCs TNFα secretion negatively correlated with the ability of moDCs to induce IL-10 producing CD4+CD25+ T cells.”

9) Several typos are present in the manuscript such as: "DC-SIGN/CD20916" (line 553) instead of "DC-SIGN/CD209", "DCsand" (line 589) instead of "DCs and", etc. Extensive proof reading is required.

As requested by the reviewer, we have corrected the mentioned mistakes and we read the entire text. We checked the consistency of the text and figures as well as the text were edited in the revised manuscript.

Reviewer #2: 

The flow cytometry data still needs to be improved.

In figure 2 : it is still not clear how the authors define the positive population for the different phenopytpic markers. If the positive population is determined according to the unlabelled cells, the mean should be higher than 80% (and not 50% as shown in the graphs).

Unlabeled cells are represented by empty histograms in Figure 2, the mean of the positive population exceeds 80% of this in all cases. The gray histograms representing RPMI-DCs provided the basis for comparison. However, for the investigated markers, in most cases, a population expressing the given molecule at a higher level (high) appeared, the percentage of which was presented in the bar charts. Comparison with only unstained or control samples or plotting the MFI would have masked the presence of cells highly expressing the markers. For complex and more objective interpretation of data, MFI values are presented on S5 Fig.

In figure s4 : a representative facs plot should be shown for CD163 and CD206.

As the Reviewer recommended, S4_Fig was completed with representative FACS results showing the expression levels of CD206 and CD163 markers. This Figure is re-uploaded to the website.

The new figure information is labelled in the text (Figure legend of S4_Fig):

“Histograms show one of the three independent experiments.”

In figure 4 : a negative control (monocyte-derived cells + Bacteria at 4°C) is lacking. The fluorescence measured should correspond to binding of the cell surface instead of phagocytosis.

We are grateful to the Reviewer for this comment or perception. As the Reviewer asked, we performed a further experimental analysis. These results are shown on Fig4 which is already uploaded to the website. 

The Figure legend is completed with the description of the new measurement:

“In vitro differentiated four-day monocyte-derived cells were coincubated with mCherry-labeled L. casei for 4h at a ratio of 1:10 at 37 °C and at 4 °C as a control. Phagocytic activity of moDCs was measured by detecting mCherry fluorescent protein-positive moDCs with flow cytometry after 4h. Diagram shows the mean plus SD of seven (37 °C) or three (4 °C) independent experiments. Boxplot shows one representative sample from the seven independent experiments.”

---

## [Decision Letter · Decision Letter 2]

22 Aug 2022

Comprehensive analysis of different tumor cell-line produced soluble mediators on the differentiation and functional properties of monocyte-derived dendritic cells

PONE-D-21-38603R2

Dear Dr. Mázló,

We’re pleased to inform you that your manuscript has been judged scientifically suitable for publication and will be formally accepted for publication once it meets all outstanding technical requirements.

Your manuscript is very much improved. However, there is still numerous mistakes (typos etc...) such as :

Line 599 The following sentence is unintelligible : “These observations corroborate the importance the simultaneous consideration of variables”

Line 610 : “but it is quite differ from the other melanomas” instead of

“but it is quite different from the other melanomas”

line 618 : “In oppose to this in the group of moDCs…” instead of

In contrast to this  

Name misspelled line 614 : Johson instead of Johnson (49).

I recommand you to have your manuscript edited if possible by a native english speaker.

Kind regards,

Jean Kanellopoulos, M.D., Ph.D.

Academic Editor

PLOS ONE

Additional Editor Comments (optional):

Reviewers' comments:

Reviewer's Responses to Questions

**Comments to the Author**

1. If the authors have adequately addressed your comments raised in a previous round of review and you feel that this manuscript is now acceptable for publication, you may indicate that here to bypass the “Comments to the Author” section, enter your conflict of interest statement in the “Confidential to Editor” section, and submit your "Accept" recommendation.

Reviewer #2: All comments have been addressed

2. Is the manuscript technically sound, and do the data support the conclusions?

Reviewer #2: (No Response)

3. Has the statistical analysis been performed appropriately and rigorously? 

Reviewer #2: (No Response)

4. Have the authors made all data underlying the findings in their manuscript fully available?

Reviewer #2: (No Response)

5. Is the manuscript presented in an intelligible fashion and written in standard English?

Reviewer #2: (No Response)

6. Review Comments to the Author

Reviewer #2: (No Response)

7. PLOS authors have the option to publish the peer review history of their article (what does this mean?). If published, this will include your full peer review and any attached files.

Reviewer #2: No

---

## [Editor Report · Acceptance letter]

26 Sep 2022

PONE-D-21-38603R2 

Comprehensive analysis of different tumor cell-line produced soluble mediators on the differentiation and functional properties of monocyte-derived dendritic cells 

Dear Dr. Mázló:

I'm pleased to inform you that your manuscript has been deemed suitable for publication in PLOS ONE. Congratulations! Your manuscript is now with our production department. 

Kind regards, 

on behalf of

Dr. Jean Kanellopoulos 

Academic Editor

PLOS ONE